# Structure of the human ATM kinase and mechanism of Nbs1 binding

**Christopher Warren[1], Nikola P Pavletich[1,2]\***

[1]Structural Biology Program, Memorial Sloan Kettering Cancer Center, New York, United States; [2]Howard Hughes Medical Institute, Memorial Sloan Kettering Cancer Center, New York, United States

**Abstract** DNA double-strand breaks (DSBs) can lead to mutations, chromosomal rearrangements, genome instability, and cancer. Central to the sensing of DSBs is the ATM (Ataxia-telangiectasia mutated) kinase, which belongs to the phosphatidylinositol 3-kinase-related protein kinase (PIKK) family. In response to DSBs, ATM is activated by the MRN (Mre11-Rad50-Nbs1) protein complex through a poorly understood process that also requires double-stranded DNA. Previous studies indicate that the FxF/Y motif of Nbs1 directly binds to ATM, and is required to retain active ATM at sites of DNA damage. Here, we report the 2.5 Å resolution cryo-EM structures of human ATM and its complex with the Nbs1 FxF/Y motif. In keeping with previous structures of ATM and its yeast homolog Tel1, the dimeric human ATM kinase adopts a symmetric, butterfly-shaped structure. The conformation of the ATM kinase domain is most similar to the inactive states of other PIKKs, suggesting that activation may involve an analogous realigning of the N and C lobes along with relieving the blockage of the substrate-binding site. We also show that the Nbs1 FxF/Y motif binds to a conserved hydrophobic cleft within the Spiral domain of ATM, suggesting an allosteric mechanism of activation. We evaluate the importance of these structural findings with mutagenesis and biochemical assays.

**\*For correspondence:**
pavletin@mskcc.org

**Competing interest:** The authors declare that no competing interests exist.

## Editor's evaluation

This manuscript is of broad interest to the DNA-repair and structural biology field. The paper describes new insights into the interaction between ATM and Nsb1, proteins central to repairing DNA double-strand breaks in humans. Overall, the structural cryo-electron microscopy data is solid and the data well analyzed and presented with key claims directly related to and supporting previous known findings.

## Introduction

Ataxia-telangiectasia mutated (ATM) is a large protein kinase with a central role in the cellular response to DNA double-strand breaks (DSBs) and related genotoxic stress (**Shiloh, 2006**). Mutations in ATM are responsible for Ataxia-telangiectasia (AT), which is a rare, autosomal recessive disorder characterized by cerebellar degeneration, immunodeficiency, sensitivity to radiation, and cancer susceptibility (**Savitsky et al., 1995**).

ATM, which belongs to the phosphatidylinositol 3-kinase-like protein kinase (PIKK) family (**Savitsky et al., 1995**), is essential for the sensing of DSBs during the cell cycle. It functions in association with the MRN protein complex consisting of the Mre11, Rad50, and Nbs1 proteins (**Syed and Tainer, 2018**). MRN contributes to the localization of ATM to DSBs, and together with double-stranded DNA (dsDNA), it activates ATM as a protein kinase. ATM then phosphorylates a wide range of downstream effector proteins, such as p53, Chk2, Brca2, and H2A.X, leading to the activation of cell cycle

checkpoints and homology-directed repair (HDR) (*Dinkelmann et al., 2009*; *Shibata et al., 2014*; *Paull, 2015*; *Uziel et al., 2003*; *Lee and Paull, 2004*; *Lee and Paull, 2005*). Mutations in all three members of the MRN complex cause disorders that are phenotypically similar to AT (*Stewart et al., 1999*; *Digweed and Sperling, 2004*; *Ragamin et al., 2020*).

The MRN complex rapidly associates with DNA ends upon DSB formation, and it is a known component of ionizing radiation-induced foci (IRIF) (*Lukas et al., 2004*; *Haince et al., 2008*). The Mre11 subunit has endonuclease and exonuclease activities that are implicated in DNA end processing (*Hoa et al., 2016*; *Myler et al., 2017*), although nuclease-inactive mutants in yeast have only mild defects in responding to ionizing radiation (IR) (*Symington and Gautier, 2011*). Rad50 is a member of the structural maintenance of chromatin (SMC) family of proteins. It contains an ATPase domain, a long antiparallel coiled coil, and a zinc hook domain that is thought to tether multiple MRN complexes (*de Jager et al., 2001*; *Hopfner et al., 2002*; *Moreno-Herrero et al., 2005*). Nbs1 contains N-terminal FHA and BRCT domains that are required for association with phosphorylated proteins such as H2A.X, CtBP-interacting protein (CtIP), and mediator of DNA damage checkpoint protein 1 (Mdc1) at IRIF (*Kobayashi et al., 2002*; *Spycher et al., 2008*). While the C-terminal half of Nbs1 is largely disordered *Williams et al., 2009* it contains short sequence motifs for binding to Mre11 and ATM (*Schiller et al., 2012*; *You et al., 2005*).

The activation of ATM by MRN is incompletely understood. The yeast homolog of ATM, Tel1, can associate with each of the individual subunits of the corresponding Mre11-Rad50-Xrs2 complex (*Hailemariam et al., 2019*). In the yeast system, Xrs2 is required for Tel1 recruitment to DSBs in vivo (*Oh et al., 2016*; *Oh et al., 2018*), but in vitro Tel1 activation appears more dependent on Rad50, as either Rad50-Mre11 or Rad50-Xrs2 but not Mre11-Xrs2 can partially activate Tel1 in the presence of DNA (*Hailemariam et al., 2019*). The ATM-binding segment of Nbs1 has been mapped to a sequence motif termed FxF/Y (residues 740–749) at the C-terminus (*You et al., 2005*). Those of Mre11 and Rad50 have not yet been identified. The Nbs1 FxF/Y motif is necessary for ATM recruitment to MRN-bound DSBs and for ATM activation in *Xenopus* extracts *You et al., 2005* and human cell lines (*Falck et al., 2005*). However, FxF/Y deletion affects only a subset of the checkpoint functions, with non-uniform effects on downstream substrate phosphorylation (*Falck et al., 2005*). In addition, fibroblasts from mice harboring an Nbs1 C-terminal deletion (Nbs1$^{\Delta C/\Delta C}$) do not display overt sensitivity to IR, although they exhibit defective intra-S phase checkpoint activation, apoptosis induction, and phosphorylation of a subset of ATM targets (*Stracker et al., 2007*). Taken together, these results suggest that the ATM-Nbs1 interaction is not strictly required to initiate ATM activation, but it is likely necessary to stabilize activated ATM at sites of DNA damage and is critical for certain ATM-mediated checkpoint functions. This may reflect the ability of the remaining MRN subunits to also interact with ATM, or additional factors within IRIFs may aid in the recruitment of ATM in vivo to partially compensate for the loss of the Nbs1 C-terminus.

Along with ATM, the PIKK family also includes ATR, DNA-PKcs, mTOR, SMG1, and TRRAP (*Imseng et al., 2018*). These mammalian PIKKs coordinate diverse cellular processes, with ATR involved in the response to collapsed replication forks, DNA-PKcs in non-homologous end joining (NHEJ), mTOR in metabolism and cellular homeostasis, SMG1 in nonsense-mediated mRNA decay, and the catalytically inactive TRRAP in scaffolding chromatin remodeling complexes (*Lovejoy and Cortez, 2009*; *Baretić and Williams, 2014*). Like ATM, these PIKK kinases are switched on by binding to their cognate activating proteins or protein complexes. The PIKKs share sequence homology across their C-terminal portion that consists of the ~700-residue FAT domain (residues 1899–2613 of ATM) and subsequent ~400-residue kinase domain (KD; residues 2614–3056 of ATM). Their N-terminal regions are divergent, and they adopt α–α solenoid structures typically consisting of two or more helical-repeat domains.

The cryo-EM structure of human ATM, based on a reconstruction of 4.4 Å to 5.7 Å resolution, showed that it forms a dimer through interactions between the FAT domains and also between the FAT and kinase domains (*Baretić et al., 2017*). The FAT-KD interactions involve a region of the KD, termed the PRD (PIKK Regulatory Domain), that sequesters the putative polypeptide substrate-binding site (*Langer et al., 2020*), as deduced from comparisons to canonical protein kinases (*Brown et al., 1999*). This arrangement was thus suggested to maintain ATM in an inactive state by inhibiting substrate binding (*Baretić et al., 2017*). The N-terminal α–α solenoids are uninvolved in dimerization (*Baretić et al., 2017*). A subsequent cryo-EM analysis reported a monomeric form at 7.8 Å resolution,

in addition to the canonical dimer at 4.3 Å (*Xiao et al., 2019*). The Tel1 homolog has been amenable to higher resolution structure determination (*Xin et al., 2019*), with recent structures of the *Chaetomium thermophilum* and *Sacharomyces cerevisiae* Tel1 reported at overall resolutions of 3.7 Å and 3.9 Å, respectively (*Jansma et al., 2020*; *Yates et al., 2020*). These Tel1 structures exhibited similar dimerization interfaces as ATM, including the region that blocks part of the putative substrate-binding site. Based on the higher resolutions of the Tel1 reconstructions, it was suggested that the kinase active site residues are in a catalytically competent organization, with the implication that the inaccessibility of the substrate-binding site is the primary mechanism of keeping ATM inactive. While Tel1 and ATM share a similar structural organization, the N-terminal ~1,800 residues preceding the FAT domain do not have detectable sequence homology.

Here, we report the cryo-EM structure of human ATM as well as the structure of ATM bound to a peptide containing the Nbs1 FxF/Y motif, both at an overall resolution of 2.5 Å. The organization of residues in the ATM catalytic cleft are very similar to those of the inactive states of mTOR and DNA-PKcs, the two PIKKs for which high-resolution structures have been reported for both the inactive and active states (*Yang et al., 2013*; *Yang et al., 2017*; *Sibanda et al., 2017*; *Chen et al., 2021b*; *Chen et al., 2021a*; *Chaplin et al., 2021*). Notably, ATM does not exhibit the conformational change that is characteristic of the active states of mTOR and DNA-PKcs kinase domains (*Yang et al., 2017*; *Sibanda et al., 2017*; *Chen et al., 2021b*). Together with biochemical data, this suggests that activation of ATM by MRN may involve a conformational change that, in addition to relieving the partial blockage of the substrate-binding site, also realigns catalytic residues.

## Results

### Structure of the human ATM dimer

FLAG-tagged ATM was purified from a stably-transfected cell line (*Figure 1—figure supplement 1A*). This preparation displays low but measurable kinase activity towards a p53 substrate peptide (*Figure 1—figure supplement 1B*). Cryo-EM samples were prepared by mixing ATM with the non-hydrolyzable ATP analog adenylyl-imidodiphosphate (AMP-PNP) and MgCl$_2$. The cryo-EM data yielded a consensus reconstruction in point group C2 that extended to an overall resolution of 2.5 Å as determined from gold standard Fourier shell correlation (FSC = 0.143) (*Supplementary file 1*, *Figure 1—figure supplement 2*). Subsequently, partial signal subtraction followed by symmetry expansion and focused refinement was used to improve the density of the N-terminal α–α solenoid regions (see Materials and methods and *Figure 1—figure supplement 3*; *Scheres, 2016*). The focused reconstructions show clear density for the majority of the side chains, allowing for the mapping of conserved residues and cancer-associated missense mutations (*Figure 1—figure supplements 3–7*). The refined model contains over 90% of the ATM residues, with the remaining residues in poorly-ordered or disordered loops. While this manuscript was being prepared, a cryo-EM structure of ATM bound to the inhibitor KU-55933 was reported at an overall resolution of 2.8 Å (*Stakyte et al., 2021*). Our structure of ATM bound to AMP-PNP is highly similar with a root mean square deviation (RMSD) of 0.9 Å based on 2,453 aligned Cα atoms.

In keeping with previous structures of ATM and Tel1 (*Baretić et al., 2017*; *Xin et al., 2019*; *Jansma et al., 2020*; *Yates et al., 2020*; *Wang et al., 2016*), ATM adopts a butterfly shaped dimer with the FAT and KD domains (hereafter FATKD) forming a dimeric body and the N-terminal α–α solenoids of ~1900 residues, previously described as Spiral and Pincer (*Baretić et al., 2017*; *Wang et al., 2016*), extending away from this body. In our structure (*Figure 1A* to C), the Spiral domain extends across residues 1–1166, the Pincer domain across residues 1167–1898, the FAT domain (named after FRAP, ATM, TRRAP) across residues 1899–2613, and the Kinase domain across residues 2614–3056. As described for other PIKKs (*Yang et al., 2013*; *Yang et al., 2017*), the Kinase domain consists of an N-terminal lobe (N lobe, residues 2614–2770) and a C-terminal lobe (C lobe, residues 2771–2957), with the catalytic cleft in between the two. The C lobe ends with the FAT C-terminal domain (FATC, residues 3027–3056) that is characteristic of the PIKK family and is absent from canonical kinases (*Yang et al., 2013*; *Yang et al., 2017*).

The N-terminal Spiral and Pincer domains, which consists mostly of HEAT repeats, are mobile relative to the FATKD dimer body. 3D classification indicated that the Spiral and Pincer domains have no preferred conformation relative to the FATKD body (*Figure 1—figure supplement 2C*).

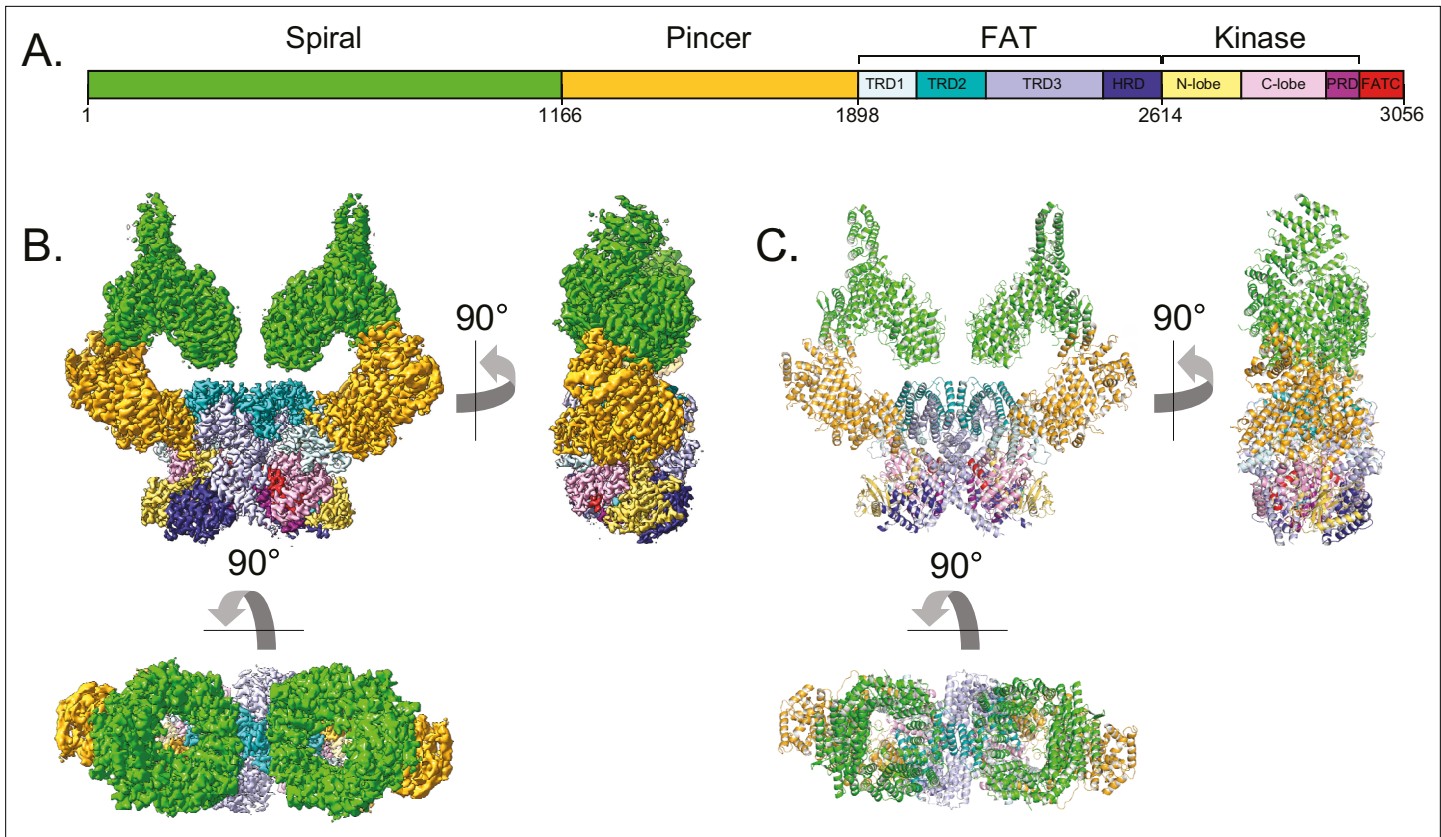

**Figure 1.** Overall structure of the human ATM kinase. (**A**) Domain map of the human ATM kinase showing Spiral (residues 1–1166), Pincer (1167–1898), FAT (1899–2613), Kinase (2614–3026), and FATC (3027–3056) domains. FAT domain is further divided into TRD1 (1899–2025), TRD2 (2026–2192), TRD3 (2193–2479), and HRD (2480–2613) subdomains. Kinase domain is further divided into N-lobe (2614–2770), C-lobe (2771–2957), and PRD (2958–3026) subdomains. (**B**) Composite cryo-EM density map of ATM kinase colored by approximate domain location. AMP-PNP molecule and $Mg^{2+}$ ion in the active site colored cyan and gray, respectively. (**C**) Structure of the overall ATM dimer colored by domain.

The online version of this article includes the following source data and figure supplement(s) for figure 1:

**Figure supplement 1.** Purification and basal activity of ATM kinase.

**Figure supplement 1—source data 1.** *Figure 1—figure supplement 1A* SDS-PAGE of purified ATM kinase.

**Figure supplement 2.** Cryo-EM data processing of the apo ATM sample.

**Figure supplement 3.** Partial signal subtraction, symmetry expansion, and focused refinement procedure for ATM.

**Figure supplement 4.** Secondary structure and sequence conservation of human ATM.

**Figure supplement 5.** Positions and frequencies of cancer-associated ATM missense mutations.

**Figure supplement 6.** Flexibility of the ATM spiral and pincer domains.

**Figure supplement 7.** Surface properties of the ATM protomer.

The most 'open' and 'closed' subclasses were each refined to an overall resolution of 2.8 Å as determined from the gold standard FSC (*Figure 1—figure supplement 2C and E*, *Video 1*). In the most open conformation, the N-terminal tips of the Spiral domains are separated by ~134 Å, while these same regions are separated by ~125 Å in the most closed conformation (*Figure 1—figure supplement 6A-D*). The mobility originates in part from flexibility within the Pincer domain, which forms the elbow-like structure of the α-solenoid arm that extends from the N-terminus to the start of the FAT domain. This flexibility is associated with the solenoid arm adopting a continuum of positions relative to the FATKD. Similar, albeit more extensive flexibility has also been observed with the cryo-EM reconstruction of *C. thermophilum* Tel1 *Jansma et al., 2020*. The flexibility of the Spiral and Pincer domains does not lead to any significant conformational changes in the FATKD segment, which can be superimposed with a 0.39 Å root mean square deviation (r.m.s.d.) in the positions of

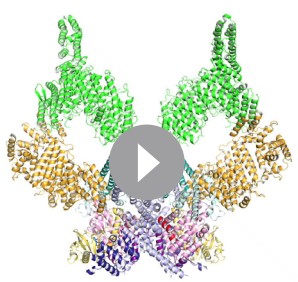

**Video 1.** Conformational flexibility of the ATM spiral and pincer domains.

https://elifesciences.org/articles/74218/figures#video1

1048 Cα atoms from the open and closed conformation structures (*Figure 1—figure supplement 6E and F*). Therefore, it is unlikely that any of these subclasses represent an intrinsically more active conformation of the ATM dimer.

The ATM FAT domain plays a central role in dimerization, which buries ~3800 Å$^2$ of surface area on each ATM protomer (*Figure 1—figure supplement 7*; *Krissinel and Henrick, 2007*). As with other PIKKs, the FAT domain consists of three Tetratricopeptide Repeat Domain subdomains (TRD1 residues 1899–2025, TRD2 residues 2026–2192, and TRD3 residues 2193–2479) followed by a HEAT-repeat subdomain (HRD residues 2480–2613) (*Figure 2A*). It adopts a 'C'-shaped structure that partially encircles the KD. In an arrangement that has been described as a C-clamp (*Yang et al., 2013*), TRD1 packs with the KD C lobe, while the HRD packs with both N and C lobes adjacent to the catalytic cleft (*Figure 2B*). This arrangement is conserved among structurally characterized PIKKs (*Imseng et al., 2018*; *Baretić and Williams, 2014*). TRD1 additionally packs with multiple regions of the Pincer domain that precedes it. Ser1981, a conserved TRD1 residue whose autophosphorylation coincides with ATM activation (*Bakkenist and Kastan, 2003*; *Pellegrini et al., 2006*), is in a nine-residue disordered loop between the fα1c and fα1d α helices (prefix 'f' denotes FAT domain helices) and is not visible in our map. However, due to the limited length of this loop and its position relative to the active site, it is unlikely that Ser1981 would be phosphorylated by either protomer in the dimeric structure. This suggests that Ser1981 is either phosphorylated by a separate ATM molecule during activation, or that a major MRN-DNA-induced structural rearrangement precedes Ser1981 autophosphorylation in the activation pathway.

TRD2, which is composed of nine α helices (fα4 to fα12), forms a substantial part of the dimer interface, with two separate regions contributing roughly one-third of the surface area buried on dimerization (~1,350 Å$^2$ of TRD2 buried per ATM protomer) (*Figure 2C*). TRD2 helices fα5 to fα8 contact the corresponding TRD2 region of the second protomer in a twofold symmetric interface (*Figure 2D*), while helices fα4 to fα5 contact TRD3 helices fα16 to fα18. The majority of these intermolecular interactions are van der Waals contacts by hydrophobic residues, along with a small hydrogen bond network and electrostatic interactions at the periphery of the hydrophobic residues. Additional buried intermolecular salt bridges between highly conserved residues Arg2032-Glu2272 and Lys2044-Glu2304 likely function to further stabilize the dimer interface.

TRD3, which is composed of 10 α helices (fα13 to fα22), accounts for the largest number of dimerization contacts and for slightly over half the surface area buried. In addition to the aforementioned TRD3-TRD2 intermolecular contacts, TRD3 also contacts the kinase domain of the second protomer. These contacts, which account for approximately one-quarter (~930 Å$^2$) of the surface area buried on dimerization, involve a C lobe surface patch that extends to the edge of the catalytic cleft (*Figure 2E and F*). The most prominent interactions are made by the TRD3 fα21-fα22 helices, which form a long coiled coil that extends across the dimer interface and packs with the C lobe near the catalytic cleft. These contacts are centered on the C lobe kα9b helix (prefix 'k' denotes kinase domain helices) that is part of the PRD (*Figure 2E*). As reported previously (*Baretić et al., 2017*; *Xin et al., 2019*; *Jansma et al., 2020*; *Yates et al., 2020*), the kα9b helix occludes part of the putative substrate-binding site, and the packing of the coiled coil against it may well stabilize this autoinhibitory conformation. A second set of contacts, made by the TRD3 helices fα19 to fα20, are centered on the FATC kα11 helix located at the end of the C lobe patch, distal from the catalytic cleft (*Figure 2F*). The FAT domain thus appears to serve multiple functions: it clamps down on the KD N and C lobes in an arrangement thought to be critical in stabilizing the inactive KD conformation of other PIKKs (*Yang et al., 2013*), it is critical for the dimerization of ATM, and it may help stabilize the kα9b conformation that occludes the putative substrate-binding site of the other protomer.

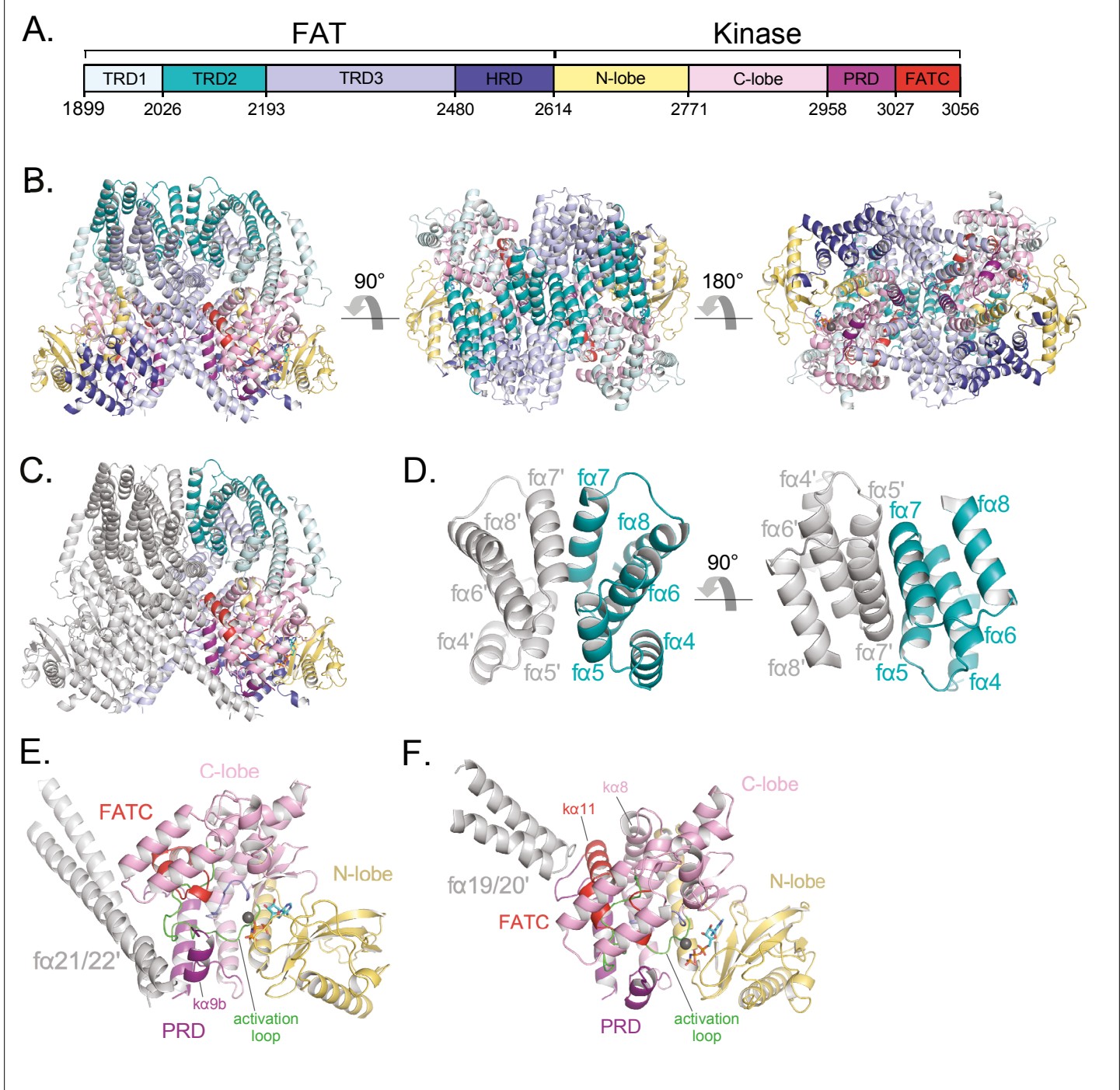

**Figure 2.** Details of the FAT domain and dimer interface of ATM. (**A**) Domain map of the FATKD colored by subdomain as in *Figure 1A*. (**B**) Structure of the FATKD colored by domain as in *Figure 1A* showing side, top and bottom views. AMP-PNP molecule and single magnesium ion colored cyan and gray, respectively. (**C**) Structure of the FATKD with one chain colored gray to highlight intermolecular contacts. (**D**) Intermolecular helical packing within the upper dimer interface by FAT helices fα4–8 located within the TRD2 subdomain. (**E**) Details of the contacts between the long coiled coil of TRD3 (fα21–22) with the kα9b of the PRD of the symmetric ATM protomer. (**F**) Contacts between fα19–20 of TRD3 with the FATC of the symmetric ATM protomer.

## Kinase domain conformation

The binding of the AMP-PNP cofactor to the catalytic cleft is overall similar to those of other PIKKs (*Baretić et al., 2017*; *Langer et al., 2020*; *Yang et al., 2013*; *Yang et al., 2017*; *Figure 3A-C*). The adenine ring is sandwiched between hydrophobic or aromatic residues from the N lobe (Leu2715, Leu2767, and Trp2769) and C lobe (Leu2877 and Ile2889), whereas the N6 and N1 groups hydrogen bond to backbone carbonyl and amide groups of Glu2768 and Cys2770, respectively (*Figure 3B and C*). Although the involvement of these residues is conserved among PIKKs, the precise position and orientation of the purine group relative to the C lobe exhibits some variation (up to ~1.5 Å and ~20°) among PIKK structures (*Langer et al., 2020*; *Yang et al., 2017*). Similarly, the conformations of the AMP-PNP ribose and phosphate groups in the ATM structure are within the range of variability among PIKK structures, albeit a wider range than that of the purine group. This is possibly due to the use of different ATP analogs, and in the case of mTOR due to conformational change between the inactive and active states (*Yang et al., 2017*).

The phosphate groups interact with the N lobe directly, through a contact between Lys2717 and the α phosphate group, and with the C lobe indirectly, through the $Mg^{2+}$ cofactor. The position of Lys2717 is equivalent to a critical lysine residue in canonical kinases, where it is thought to orient the γ phosphate group for phosphotransfer (*Yang et al., 2013*; *Knighton et al., 1991*). Whether the lysine residue has an analogous role in PIKKs is not yet known. In addition to this lysine residue, canonical kinases have an N lobe β hairpin, rich in glycine residues, that packs against the phosphate groups when both the ATP and peptide substrates are assembled (*Knighton et al., 1991*). The corresponding region in PIKKs is more variable in sequence. In ATM, it contains Gly2694 and Gly2695, with the latter's backbone amide being ~4 Å away from the β phosphate group. This glycine-rich loop also contains Val2696 and Asn2697 that, together with Tyr2969 on kα9b, appear to occlude the γ phosphate (*Figure 3D*). The $Mg^{2+}$ ion, which contacts the AMP-PNP α phosphate group, is coordinated by Asn2875 and Asp2889 of the C lobe – interactions conserved among both PIKKs and canonical protein kinases (*Imseng et al., 2018*; *Xin et al., 2019*; *Jansma et al., 2020*; *Yates et al., 2020*; *Yang et al., 2013*). Our maps show only a low level of density for a second $Mg^{2+}$ ion thought to be involved in phosphoryl transfer by canonical kinases (*Bao et al., 2011*; *Jacobsen et al., 2012*). The second Mg is often poorly ordered in canonical kinases as well as PIKKs (*Xin et al., 2019*; *Jansma et al., 2020*; *Yates et al., 2020*).

On the C lobe, the magnesium ligands and other critical catalytic residues map to two loops, named catalytic loop (residues 2866–2875) and activation loop (residues 2888–2911) by analogy to canonical kinases (*Yang et al., 2013*). As with previous studies of ATM and Tel1, these loops are well ordered and their catalytic residues show clear density in our maps. The catalytic loop contains Asp2870, Asn2875 (a $Mg^{2+}$ ligand) and His2872. Asp2870 acts as the catalytic base to deprotonate and likely orient the hydroxyl group of the incoming substrate (*Vadas et al., 2011*) and His2872 is thought to stabilize the transition state of the phosphoryl transfer reaction (*Yang et al., 2013*). The activation loop is named for its conformational change in the activation of canonical kinases, where it makes up part of the polypeptide substrate-binding site. Previous studies showed that the activation loop of ATM/Tel1 packs with the kα9b helix of the PRD, and by analogy to canonical kinases it was suggested that the kα9b helix may block substrate binding. This was recently confirmed by the cryo-EM analysis of the PIKK Smg1, which, like ATM, is specific for a glutamine residue in the position ($P_{+1}$) after the serine/threonine phosphorylation site. In the Smg1-UPF1 substrate structure (*Langer et al., 2020*), the glutamine side chain of the peptide substrate makes a pair of hydrogen bonds to backbone amide and carbonyl groups on the activation loop. This interaction is effectively mimicked in our ATM structure by the side chain of Gln2971 on the kα9b helix of ATM (*Figure 3—figure supplement 1*).

It has been suggested that the Tel1/ATM active site residues are in a catalytically competent conformation, and thus the occlusion of the substrate-binding site would be the main mechanism of keeping the kinase autoinhibited (*Baretić et al., 2017*; *Xin et al., 2019*; *Jansma et al., 2020*; *Yates et al., 2020*). However, studies of mTOR have shown its activation involves a conformational change in the FAT domain, which in turn allows the KD N and C lobes to move relative to each other. This results in the realignment of catalytic residues on the N and C lobes relative to each other, bringing them into the correct register for efficient catalysis (*Yang et al., 2017*). The recently reported active-state structure of DNA-PKcs recapitulates the N-C lobe conformational change on activation (*Chen et al.,*

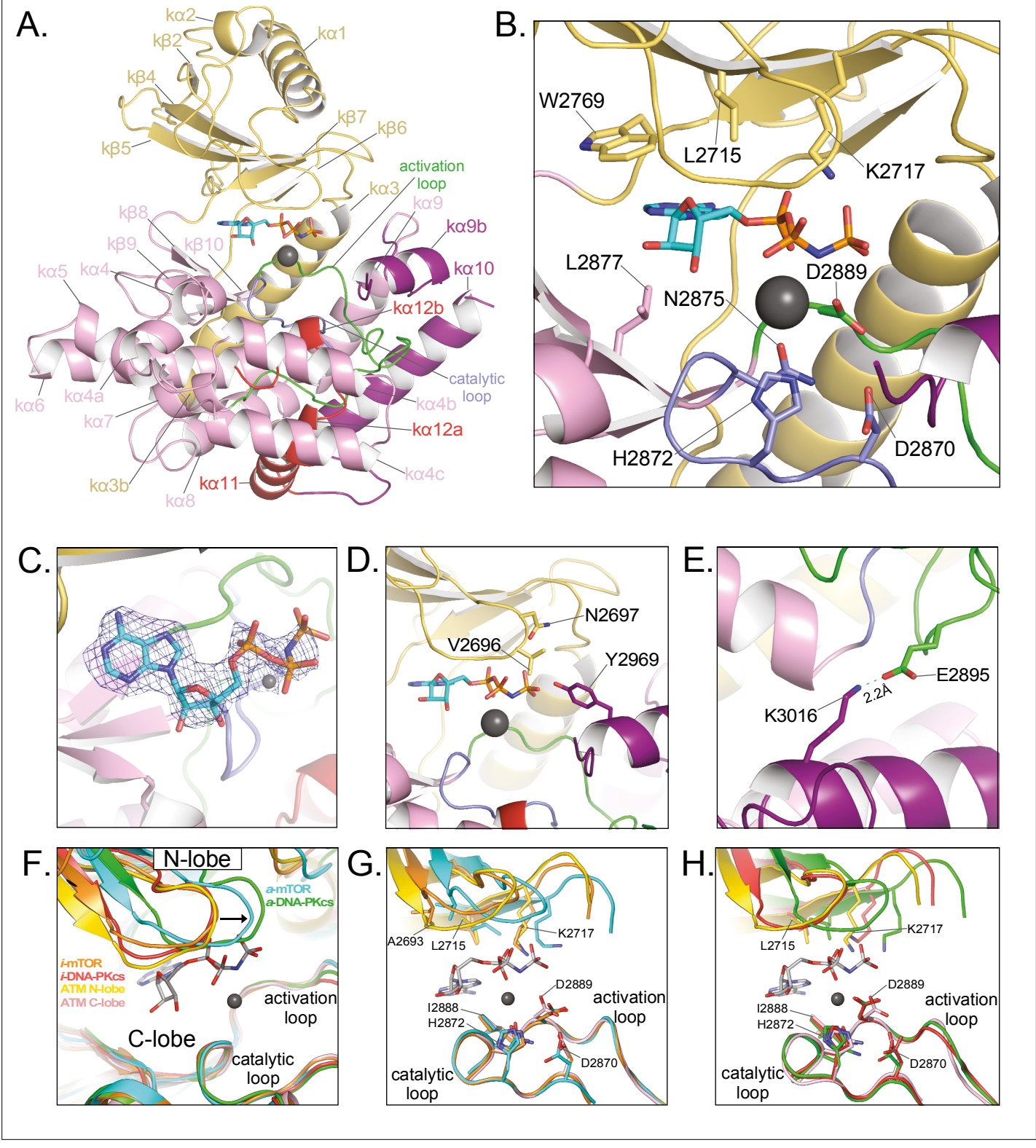

**Figure 3.** Details of the kinase domain and active site of ATM. (**A**) Structure of the Kinase, PRD, and FATC domains with secondary structure elements labeled. Also highlighted are the locations of the catalytic loop (residues 2866–2875, blue) and activation loop (residues 2888–2911, green). AMP-PNP and a single magnesium ion in the active site colored cyan and gray, respectively. (**B**) Details of the interactions within the active site. Critical residues within the N lobe, C lobe, catalytic and activation loops are shown as sticks and labeled. (**C**) Electron density around AMP-PNP and Mg²⁺ cofactors contoured to 5σ. (**D**) Steric occlusion of the γ phosphate of AMP-PNP by residues V2696 and N2697 on the N lobe and Y2969 on kα9b of the PRD.

*Figure 3 continued on next page*

*Figure 3 continued*

(**E**) Salt-bridge formed between K3016 (PRD, kα10) to E2895 (activation loop). K3016 is acetylated during the DNA-damage response and correlates with active ATM kinase. (**F**) Alignment of the Kinase domain of ATM with those of mTOR and DNA-PKcs in the inactive and active states. Structures are aligned along their corresponding catalytic and activation loops. ATM N and C lobes are colored yellow and pink, respectively. Inactive and active mTOR are colored orange and cyan, respectively. Inactive and active DNA-PKcs are colored red and green, respectively. Arrow indicates movement of the N lobe relative to the C lobe coincident with mTOR and DNA-PKcs activation. (**G**) Positions of catalytically important residues relative to those of mTOR in the inactive and active states. (**H**) Positions of catalytically important residues relative to those of DNA-PKcs in the inactive and active states.

The online version of this article includes the following figure supplement(s) for figure 3:

**Figure supplement 1.** Blockage of the substrate-binding site.

*2021b*), raising the possibility that this is a common activation mechanism of PIKKs, irrespective of their distinct activators and N-terminal solenoid structures to which these activators bind.

Thus, to help evaluate whether the ATM active site residues are in a catalytically competent conformation, we superimposed the KD domains of ATM, mTOR and DNA-PKcs, the latter two in both their active and inactive conformations, by aligning their C lobe catalytic and activation loops. In this superposition (*Figure 3F*), the ATM N lobe is positioned remarkably similar to those of inactive mTOR and inactive DNA-PKcs, with the three N lobes forming a tightly clustered set clearly distinct from the cluster of active mTOR and active DNA-PKcs N lobes. The positions of the N lobe hairpin and other ATP-interacting residues relative to the C lobe catalytic and magnesium-coordinating residues are much closer to those of the inactive mTOR and DNA-PKcs structures compared the active ones (*Figure 3G and H*). This positioning thus suggests that in addition to relieving the blockage of the substrate-binding site, the activation of ATM may well involve a conformational change within the FAT domain and the associated change in the relative orientation of the N and C lobes of the kinase domain.

## Structure of ATM bound to the C-terminus of Nbs1

Previous studies indicated that the Xrs2 C-terminal FxF/Y motif and the acidic region that immediately precedes it bind to two Tel1 regions spanning the Spiral and Pincer domains near the elbow (*You et al., 2005*; *Yates et al., 2020*; *Figure 4A*). As the Nbs1 C-terminal half is unstructured or loosely folded (*Williams et al., 2009*), we made cryo-EM grids using a 28-residue peptide (207 μM) that encompasses the acidic region and FxF/Y motif of human Nbs1 (residues 727–754, hereafter referred to as Nc28) and human ATM (1.2 **μ**M).

The cryo-EM data yielded a consensus reconstruction in point group C2 that extended to 2.5 Å resolution as determined from the gold standard FSC (*Figure 4—figure supplement 1*). The initial map had additional density, absent in the apo ATM (*Figure 4—figure supplement 1D*), at the ATM Spiral domain. Partial signal subtraction and symmetry expansion procedures followed by iterative focused 3D classifications identified 75% of the particles that had the extra density (481,066 particles; *Figure 1—figure supplement 3A* and *Figure 4—figure supplement 1C*). After focused refinements of the ATM Spiral domain, we built a 10-residue Nbs1 segment ([740]ADDLFRYNPY[749]) into the improved density guided by the unambiguous density of the side chains for Phe744 and Tyr746. According to local resolution estimation, the focused reconstruction has a resolution better than ~3.5 Å in the central portions of the peptide (*Figure 4B*, *Figure 4—figure supplement 1G*). There is no interpretable density for residues 727–739 preceding the built segment or residues 750–754 after. We note that in contrast to previous yeast two-hybrid assays indicating that the C-terminus of Xrs2 interacts with both the Spiral and Pincer domains of Tel1 (*You et al., 2005*), we find no evidence of the ATM Pincer domain being involved in binding to the Nbs1 peptide.

The Nbs1 peptide adopts an overall extended conformation except for one turn of a $3_{10}$ helix in the middle (residues Asp742 to Arg745; *Figure 4C and D*). It binds to a hydrophobic groove between two helical repeats formed by helices sα39 to sα42 (denoted 's' for Spiral domain helices). The most extensive ATM contacts are made by Nbs1 Leu743 and Phe744, which pack together preceding the $3_{10}$ helix. Phe744 inserts deepest into the hydrophobic groove and makes van der Waals contacts to the side chains of Ser978, Arg981, Cys987, Val1021, Ala1024, Phe1025, Leu1028, and Tyr1034 of ATM (*Figure 4C and D*). Leu743, which is closer to the solvent exposed surface of the ATM groove, packs with Ala1024, Leu1028, and His1027 (*Figure 4C and D*). Of the two acidic residues that precede the Leu743-Phe744 pair, Asp741 hydrogen bonds with Ser978 and Arg981, while Asp742 is solvent

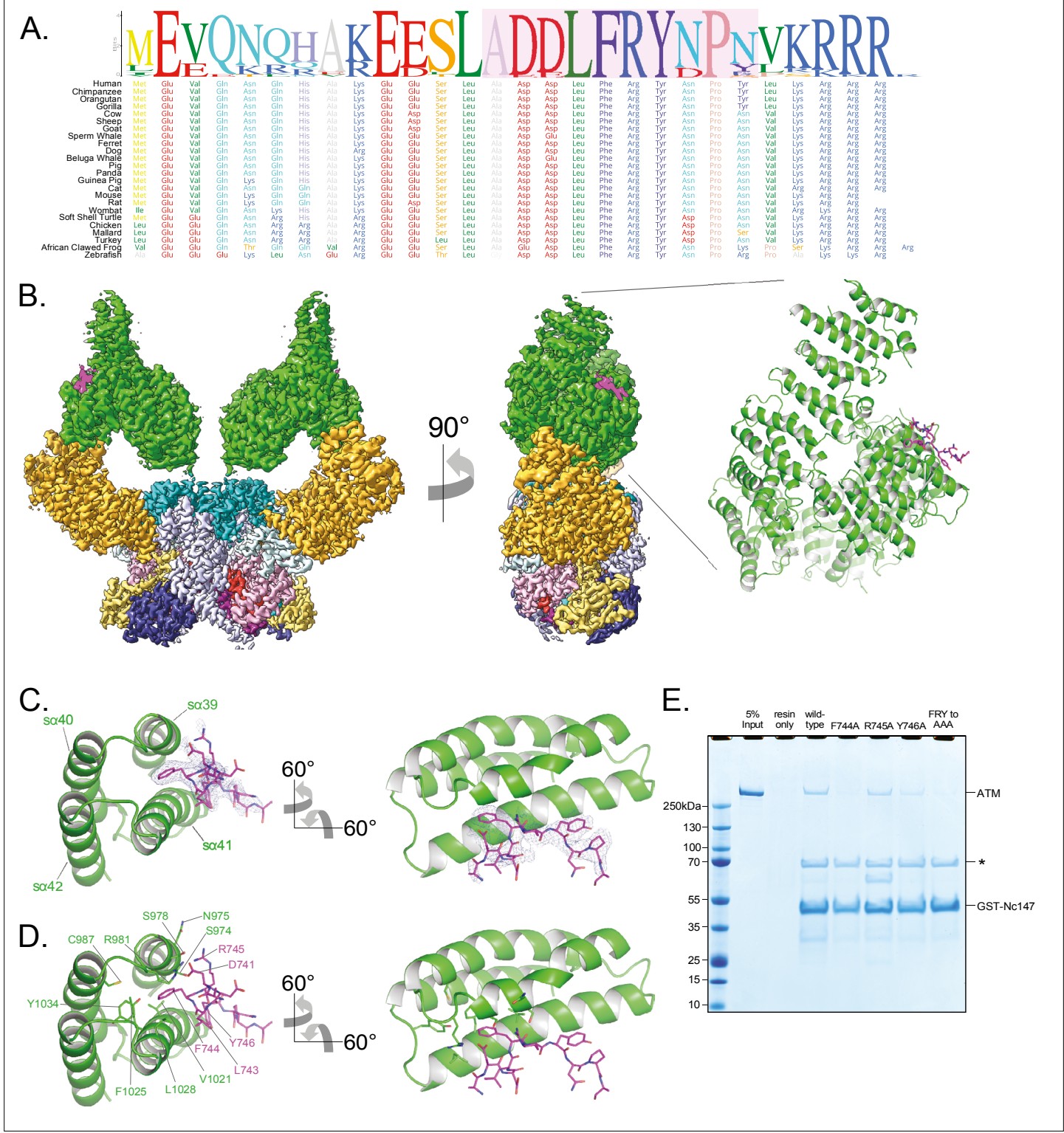

**Figure 4.** Structure of ATM bound to the C-terminus of Nbs1. (**A**) Alignment of the Nbs1 C-terminal 28 residues (Nc28) from 24 vertebrate species showing the conservation of the C-terminus and FxF/Y motif. Portion of Nc28 visible in the cryo-EM map is highlighted in magenta. (**B**) Location of the Nc28 bound to the ATM Spiral domain in the composite map (left) and structure (right). Nc28 is colored magenta. (**C**) Zoom of the location of the Nc28 peptide showing Phe744 inserted into a hydrophobic groove made up of ATM helices sα39 to sα42. Nc28 peptide density is contoured to 4σ. (**D**) Details of the interaction between Nc28 and the ATM Spiral, showing Phe744 inserted into a hydrophobic pocket created by ATM residues Arg981, Cys987, Val1021, Ala1024, Phe1025, Leu1028, and Tyr1034. Arg745 of Nbs1 makes electrostatic contacts with ATM Asn975 and Ser978. Tyr746 of Nbs1

*Figure 4 continued on next page*

*Figure 4 continued*

packs against the sα41. Nbs1 L743 makes contacts with an adjacent hydrophobic cleft created by ATM residues Leu1028 and His1064. Nbs1 Asp741 makes electrostatic and H-bonding contacts with ATM residues Arg981 and Ser978, respectively. (**E**) ATM pull-down assay using GST-tagged Nbs1 C-terminal 147 residues (GST-Nc147) as bait. Lanes are labeled at the top of the image, and bands are labeled on the right. Asterisk indicates Hsp70/DnaK contaminant that co-purifies with all GST-Nc147 preparations.

The online version of this article includes the following source data and figure supplement(s) for figure 4:

Source data 1. *Figure 4E* Nbs1 pull-down assay.

Figure supplement 1. Cryo-EM data processing of the ATM-Nc28 sample.

Figure supplement 2. Structural assessment of ATM bound to the Nc28 peptide.

Figure supplement 2—source data 1. *Figure 4—figure supplement 2C* ATM kinase assay with DNA and Nc28.

---

exposed and uninvolved in ATM contacts (*Figure 4C and D*). Arg745, the last residue of the $3_{10}$ helix, has weaker side chain density, although its guanidinum group is positioned between the side chains of Asn975 and Ser978 (*Figure 4C and D*). Tyr746 is the last Nbs1 residue that contacts ATM. Its side chain, which is stabilized by intramolecular stacking with Pro748, packs with side chain and backbone groups of Gly1016, Gln1017, and Thr1020 of ATM (*Figure 4C and D*).

The ATM residues that make up the Nbs1 binding site, and in particular those at the Leu743-Phe744 binding pocket, are highly conserved compared to the rest of Spiral domain residues (*Figure 1—figure supplement 4* and *Figure 4—figure supplement 2A*). In addition, many of these ATM residues have been found mutated in cancer (*Figure 1—figure supplement 5D and E*). Ser978 is a hotspot for cancer-associated missense mutations and the structure suggests that the S978P mutation would disrupt the sα39 helix that forms part of the Nbs1-binding site, whereas S978A and S978Y would eliminate the interactions of Ser978 with Asp741 of Nbs1 (*Figure 1—figure supplement 5D and E*). Other cancer-associated missense mutations at the Nbs1 binding groove occur at lower frequencies. The structure suggests that the S974F mutation, in a region of limited solvent accessibility due to the Nc28 peptide, would likely cause a steric clash either with Arg745 of Nbs1 or other ATM backbone and side chain groups; R981C/H would eliminate the contact to Asp 741 of Nbs1; C987Y/W and F1025S/L, which map to a local hydrophobic core behind the sα39 and sα42 helices, would disrupt the structural integrity of the binding site of Nbs1 Phe744 and its vicinity. The abundance of mutations at the Nbs1-binding site suggest that the ATM-Nc28 interaction is functionally important for the ATM-mediated DNA damage response. Reported mutations in Nbs1 are fewer, but include the F744L mutation that would disrupt the most critical portion of the ATM-Nc28 interaction (*Figure 1—figure supplement 5F*).

The ATM structural elements at the Nbs1 binding site, including the hydrophobic pocket and adjacent cleft, are nearly identical in both the apo and Nc28-bound structures, indicating that Nc28 binding does not induce any noticeable structural rearrangements to this region (*Figure 4—figure supplement 1D*). Additionally, no significant changes to the structure of the kinase domain were observed when comparing the unbound and Nc28-bound states (*Figure 4—figure supplement 2D*). Consistent with these observations and with reports that ATM activation requires all three members of the MRN complex and dsDNA (*Lee and Paull, 2005*), we observed no increase in ATM catalytic activity in steady-state kinase assays containing high concentrations of this peptide (*Figure 4—figure supplement 2C*). Finally, although previous studies suggest that the Nbs1 C-terminus may interact with the ATM FATC domain (*Xin et al., 2019*; *Ogi et al., 2015*), we observe no additional electron density within this region of ATM despite our sample containing a large molar excess of the Nc28 peptide.

The N-terminal acidic half of the peptide is not visible in our map. However, the orientation of the visible portion indicates that it likely extends into a basic patch created by ATM residues Arg981, Arg982, and Lys1033, raising the possibility that the acidic residue clusters (Glu728, Glu736, and Glu737) contribute to binding through long-range electrostatic interactions (*Figure 4—figure supplement 2B*). Alternatively, the acidic residues may engage in ATM interactions in the context of the intact MRN complex. Similarly, while the C-terminal 4 residues of Nbs1 ([750]LKRRR[754]) are also not visible, they would be in the vicinity of an acidic patch of ATM (Glu964, Asp965, Asp1007, Glu1009, and Asp1013; *Figure 4—figure supplement 2B*).

To evaluate the relative contributions of the ATM-Nbs1 contacts, we made several mutations in the FxF/Y motif and tested the ability of these mutant peptides to bind to ATM in a GST pull-down assay. Four mutations (F744A, R745A, Y746A, and [744]FRY[746] to [744]AAA[746]) were introduced into a GST-tagged polypeptide corresponding to the C-terminal 147 residues of Nbs1 (GST-Nc147). While the wild-type GST-Nc147 and the R745A mutant both enriched for ATM in pull-down assays, all other mutations disrupted this interaction at least partially (*Figure 4E*). F744A and FRY to AAA mutations appeared to disrupt the interaction almost completely, whereas Y746A showed an intermediate effect. These results indicate that Nbs1 Phe744 is central to ATM binding, Tyr746 clearly contributes, while Arg745 makes at most a minor contribution in our assay.

The Mre11 binding motif of Nbs1 (residues 682–693) is located approximately 60 residues N-terminal to the FxF/Y motif, indicating that the MRN complex likely binds along the Spiral and/or Pincer domains of ATM. Interestingly, we mapped multiple patches of conserved residues to the inner portion of the Spiral domain α-solenoid, which may represent binding sites for MRN-dsDNA or other effector proteins (*Figure 1—figure supplement 7B*). Notably, the Nbs1 interaction site is located over 100 Å away from the active site. This is reminiscent of mTOR and DNA-PKcs, whose activators, respectively Rheb and Ku-DNA, bind far from the KD and activate their respective kinases allosterically[42,44]. This mechanism would suggest that the full MRN-dsDNA complex induces structural changes that are transmitted through the Spiral and Pincer domains to the FATKD, although the Rad50 coiled coil domain may be long enough to extend from the head of the MRN complex and make direct contacts to the KD and/or FATC to promote activation.

## Biochemical analysis of MRN-mediated ATM activation and the role of the Nbs1 C-terminus

To better understand how human ATM kinase is activated and to further evaluate the role of the Nbs1 C-terminus, we expressed the human MRN complex in mammalian cells via transfection of a polycistronic vector (see Materials and methods) and purified it to near homogeneity (*Figure 5—figure supplement 1A*). Each protein contains a C-terminal FLAG tag and Mre11 harbors the H129N mutation, which abolishes nuclease activity without interfering with DNA binding and ATM activation (*Buis et al., 2008*). We first evaluated the effect of increasing concentrations of the purified MRN complex on ATM activity using a substrate peptide from p53, whose phosphorylation by ATM activates the transcription program associated with the G1/S DNA-damage checkpoint. In keeping with findings with the Tel1-MRX homologs (*Hailemariam et al., 2019*), MRN alone stimulated the ATM kinase activity only modestly, by a factor of ~4 (*Figure 5—figure supplement 1B*). The half-maximal effective concentration (EC$_{50}$) of MRN was ~27 nM, indicative of a high affinity for ATM.

We next tested linear dsDNA fragments of lengths ranging from 100 to 2000 base pairs (bp) for their ability to stimulate ATM (25 nM) in the presence of MRN (250 nM; concentration ~10 fold above its EC$_{50}$ toward ATM alone). The dsDNA fragments were added at a constant base-pair concentration of 3.85 μM (2.5 ng/μL of each fragment). dsDNA lengths greater than ~200 bp lead to maximal ATM activation, representing an additional ~25 fold increase in p53 phosphorylation relative to ATM-MRN alone, or a ~100 fold increase relative to ATM alone (*Figure 5A*, *Figure 5—figure supplement 1H*). This dsDNA length dependency for ATM-MRN activation is qualitatively similar to that reported for the Tel1-MRX complex (*Hailemariam et al., 2019*). We then performed a dose response analysis for dsDNA of 100, 250 and 500 bps lengths. 250 and 500 bp dsDNA fragments stimulated with very similar EC$_{50}$ values of 1.8 nM and 1.5 nM, while stimulation by 100 bp dsDNA fragment was a factor of 5 lower compared to the maximal levels of the longer DNA fragments, even at the ~1000 fold higher concentration of 2.3 μM (*Figure 5B* and *Figure 5—figure supplement 1D*). This suggests that the low-level activation of shorter dsDNA fragments is likely not due to their reduced affinity for ATM-MRN, and that structural aspects of longer DNA, such as the ability to span an extended binding site across the ATM dimer or to link distal binding sites on ATM-MRN, may be important. Stimulation by dsDNA is strictly dependent on the presence of the MRN complex, as in the absence of MRN, 350 bp dsDNA at a concentration of 25 nM, which is more than 10-fold above the EC$_{50}$ of the 250 and 500 bp DNA failed to activate (*Figure 5—figure supplement 1E and H*).

We next evaluated whether MRN-dsDNA affects the intrinsic catalytic step or peptide substrate binding. The steady state kinetic analysis of ATM phosphorylating the p53 substrate showed that MRN-dsDNA increased the catalytic step 460-fold, with $k_{cat}$ values of 0.0005 s$^{-1}$ and 0.24 s$^{-1}$ in the

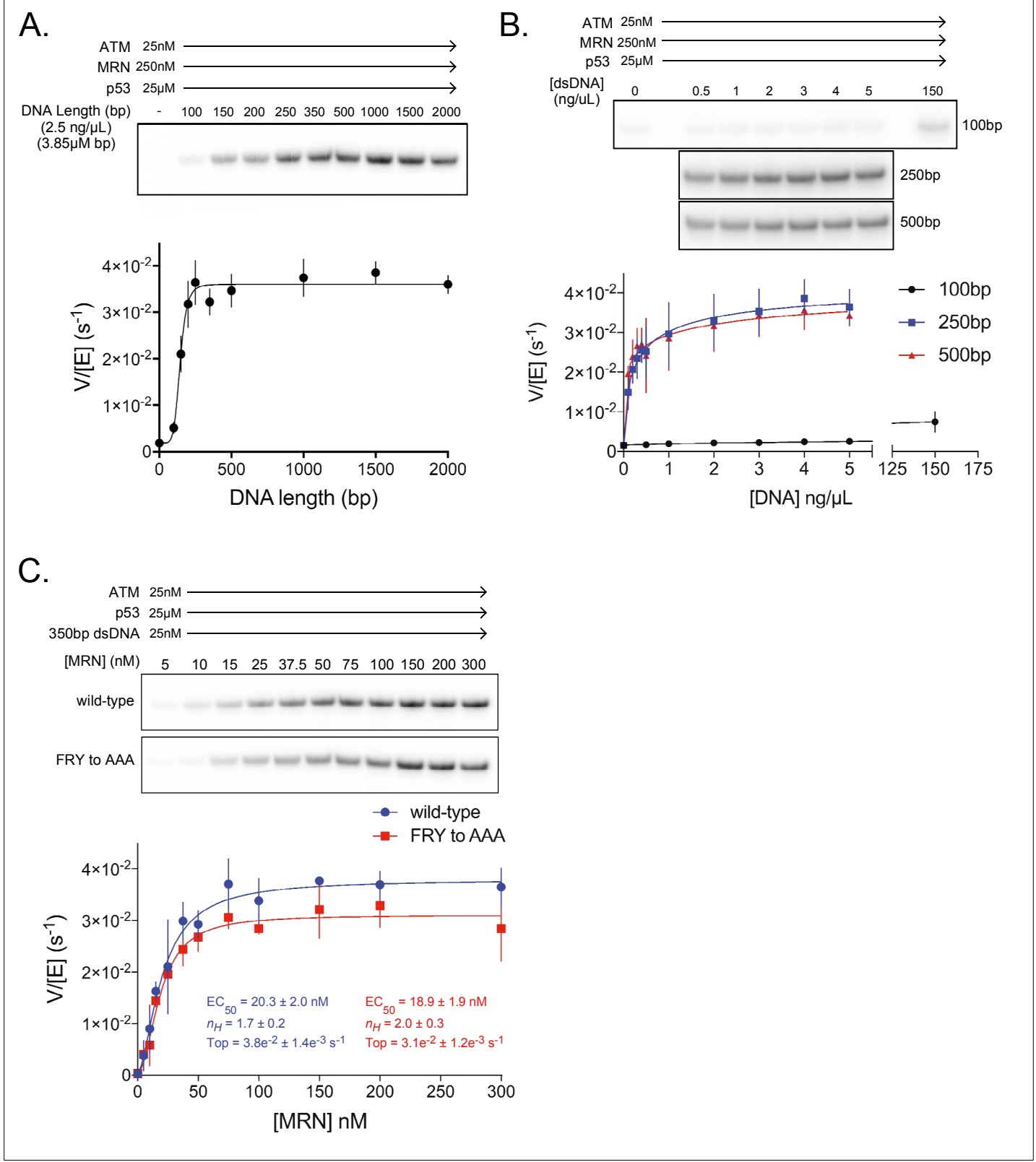

**Figure 5.** ATM activation requires MRN and long dsDNA, but not the FxF/Y motif of Nbs1. (**A**) *Top*: Steady-state ATM kinase assay using 25 nM ATM, 250 nM MRN, and 25 μM p53 substrate with 2.5 ng/μL of dsDNA of various lengths added. *Bottom*: Quantification of ATM enzymatic velocity as a function of DNA length. Three biological replicates per point. Curve fit to the $EC_{50}$ equation. (**B**) *Top*: Steady-state ATM kinase assay using 25 nM ATM, 250 nM MRN and 25 μM p53 substrate with various concentrations (0.5–5 ng/μL) of 100 bp, 250 bp, and 500 bp dsDNA fragments. 150 ng/μL 100 bp

*Figure 5 continued on next page*

*Figure 5 continued*

dsDNA is in the upper right lane. *Bottom*: Quantification of ATM enzymatic velocity as a function of DNA concentration. Three biological replicates per point, Curve fit to the EC$_{50}$ equation. (**C**) *Top*: Steady-state ATM kinase assay using 25 nM ATM, 25 nM 350 bp dsDNA, and various concentrations of wild-type or FRY to AAA mutant MRN. *Bottom*: Quantification of ATM enzymatic velocity as a function of MRN concentration. Three to six biological replicates per point. Curves fit to the EC$_{50}$ equation.

The online version of this article includes the following source data and figure supplement(s) for figure 5:

**Source data 1.** *Figure 5A* ATM kinase assay with MRN and varying DNA length.

**Figure supplement 1.** Quantitative ATM kinase assays.

**Figure supplement 1—source data 1.** *Figure 5—figure supplement 1A* SDS-PAGE of purified MRN complex.

**Figure supplement 2.** Structural conservation of the FAT domain anchor and model of FATKD activation.

absence and presence of MRN-DNA, respectively, while the $K_M$ values were unchanged within experimental error (58 µM and 67 µM; *Figure 5—figure supplement 1E* and F). Time course assays performed under single-turnover conditions also indicate that MRN directly increases the rate of the catalytic step, as opposed to a hypothetically rate-limiting product release step (*Figure 5—figure supplement 1G*).

We next made MRN containing the Nbs1 [744]FRY[746] to [744]AAA[746] mutation and titrated this mutant complex into kinase assays with saturating amounts of 350 bp dsDNA present (*Figure 5C*). In contrast to pull-down assays, we observe only a minor defect in MRN-mediated activation of ATM kinase activity with this mutant. The maximal catalytic rate of ATM with this mutant was approximately 18% lower compared to the wild-type MRN complex, although an effect on the EC$_{50}$ of activation was not discernible. We note, however, that our in vitro assay conditions, such as the saturating dsDNA concentration, may not recapitulate the demonstrated requirement for MRN recruiting ATM to DNA in vivo (*Oh et al., 2016*; *Oh et al., 2018*; *Falck et al., 2005*).

## Discussion

Our cryo-EM structure of human ATM shows an overall symmetric dimer, albeit with the Pincer and Spiral domains exhibiting some mobility arising from conformational flexibility within the Pincer domain. Nevertheless, this conformational flexibility does not translate to any noticeable structural rearrangements within the FATKD segment, nor does it break the inherent symmetry of this segment (*Figure 1—figure supplement 6*). Previous cryo-EM studies of human ATM and yeast Tel1 have observed asymmetric dimers and monomers, the latter of which was shown to have increased catalytic activity (*Baretić et al., 2017*; *Xiao et al., 2019*). In our purifications and subsequent cryo-EM structure determination, we failed to find any particles representing asymmetric dimers or monomers, also consistent with the most recent cryo-EM structures of Tel1 (*Jansma et al., 2020*; *Yates et al., 2020*). While it is not clear how the asymmetric dimers and monomers arise, it is conceivable that expression and purification conditions may play a role. Detergents have previously been demonstrated to disrupt the ATM dimeric structure and increase ATM kinase activity independent of MRN (*Lee and Paull, 2006*). We note that the report of active monomer preparations demonstrated only an approximately 10-fold increase in catalytic activity relative to the inactive dimeric state (*Xiao et al., 2019*), whereas our quantitative kinase assays show a ~100 fold activation upon MRN + long dsDNA addition (*Figure 5—figure supplement 1H*).

As noted in previous ATM and Tel1 structures (*Baretić et al., 2017*; *Xin et al., 2019*; *Jansma et al., 2020*; *Yates et al., 2020*), the apparent reason for the inhibited state of the ATM dimer is the kα9b helix packing against the putative substrate-binding site and inserting a glutamine side chain into the activation loop, at a site where a glutamine side chain from the substrate would bind (*Figure 3—figure supplement 1*; *Langer et al., 2020*). As the kα9b helix packs with the TRD3 coiled coil of the other protomer in the dimer, an interaction that possibly stabilizes the kα9b conformation, the TRD3 coiled coil was proposed to couple ATM dimerization to autoinhibition (*Baretić et al., 2017*; *Xin et al., 2019*; *Jansma et al., 2020*; *Yates et al., 2020*).

The kα9b helix is part of the PRD domain, which was first described based on a yeast two-hybrid screen to identify the part of the ATR PIKK that interacts with its activator TopBP1 (*Mordes et al., 2008*). The screen led to a minimal ATR fragment that encompasses kα9b and kα10. While scanning

mutagenesis identified an ATR kα10 residue that disrupts TopBP1-mediated activation (*Mordes et al., 2008*) it is not clear what role kα10 plays in PIKK activation. The mutation (K2589E) maps to a surface residue uninvolved in any interactions in the inactive ATR structure (*Rao et al., 2018*), and the corresponding ATM residue (Glu3007) is similarly surface-exposed and devoid of any interactions. In addition, the kα10 helix is an integral part of the PIKK C lobe structure, and it is structurally invariant among inactive and active PIKK structures. It is thus unlikely that kα10 has a significant role in conformationally regulating the kinase. In ATM, the kα9b and kα10 helices are connected by a 27-residue loop (residues 2975–3000) that is not conserved across species and is not visible in our map (*Figure 1—figure supplement 4*). An unstructured loop intervening the kα9b and kα10 helices is present in most PIKK structures reported to date, including a >1000 residue insertion in Smg1[35].

In structures of inactive DNA-PKcs, kα9b and subsequent kα9c, a helix unique to DNA-PKcs, also pack against the substrate-binding site and occlude it (*Sibanda et al., 2017*; *Chen et al., 2021b*). However, even though DNA-PKcs is glutamine directed as ATM, it does not have the equivalent of the Gln2971-activation loop interaction of ATM. Nevertheless, upon DNA-PKcs activation, the pair of kα9b and kα9c helices move to an alternate packing position on the C lobe and fully expose the substrate-binding site (*Chen et al., 2021b*). The movement of kα9b-kα9c is likely due entirely to the realignment of the N and C lobes of DNA-PKcs, as this segment only interacts with the kinase domain (DNA-PKcs does not have the equivalent of the ATM TRD3 coiled coil packing with and possibly stabilizing kα9b across a dimer interface). By contrast, in mTOR, which is not glutamine directed and instead prefers a hydrophobic residue at the $P_{+1}$ position, the kα9b helix does not block the putative substrate-binding site, nor does it change conformation on activation (*Yang et al., 2017*). Thus, while kα9b may play a role in the autoinhibition of ATM and a number of other PIKKs, this is not a conserved feature of the PIKK family.

Rather, several lines of evidence point to the realignment of the N and C lobes of the kinase domain as the key event in the activation of ATM. As discussed above, the relative orientation of the N and C lobes of ATM is remarkably similar to those of the inactive-state mTOR and DNA-PKcs structures, and distinct from those of the active-state counterparts, which cluster together (*Figure 3F*). The proper alignment of the N and C lobes is central to kinase activation, as residues critical for substrate binding and catalysis are distributed across the two lobes (*Yang et al., 2017*).

The mechanism of a PIKK activation was initially established by comparing the inactive and active states of the mTOR holoenzyme (*Yang et al., 2017*). That work pointed to the FAT domain being the key autoinhibitory element that keeps the N and C lobes and their active site residues in an unproductive configuration (*Yang et al., 2017*). Binding of the mTOR activator, the small GTPase Rheb, causes a motion of the N-heat solenoid (comparable to the ATM Spiral in its location in the primary sequence but not in its structure). The motion of the N-heat solenoid pulls and twists the FAT domain on which it is anchored, causing it to undergo an extensive conformational change. This shifts the HRD of the FAT domain away from the N lobe, allowing the N lobe to relax to a productive configuration relative to the C lobe. The inhibitory effect of the FAT domain is supported by activating mTOR mutations, which map to residues that couple the N and C lobes and the FAT domain (*Yang et al., 2017*).

The recent cryo-EM structures of activated DNA-PKcs recapitulate the FAT conformational change observed in mTOR (*Chen et al., 2021b*). DNA-PKcs activation is triggered by the N-heat solenoid binding to the dsDNA end, stabilized by the Ku70-Ku80 complex. The Ku complex interacts with DNA and the N- and M-heat DNA-PKcs domains, but it also utilizes two flexibly linked elements for additional contacts to DNA-PKcs. These may reflect an initial recruitment interaction, a role that the Nbs1 Nc28 peptide may also be involved in (*Figure 4—figure supplement 2E*). On DNA-end binding, the DNA-PKcs N-heat solenoid moves relative to M-heat. As with mTOR, N-heat is anchored on the FAT domain, and its motion allosterically induces a conformational change in the FAT domain that realigns the N and C lobes.

In both mTOR and DNA-PKcs, the N-heat solenoids that move are anchored on the FAT TRD2-TRD3 interface, at essentially the same location. And, even though their overall N-heat solenoids are structurally distinct, they both use 4-helix bundle followed by a loop to interact with their respective FAT domains. Our ATM structure reveals a remarkably similar interface between the FAT domain and the Pincer (equivalent to M-heat of mTOR and DNA-PKcs) from the same protomer. The similarities include an identical TRD2-TRD3 site of the FAT domain and a 4-helix bundle and loop element from the Pincer that binds to it, in essentially the same configuration as those of mTOR and DNA-PKcs

(*Figure 5—figure supplement 2A*). Based on this structural conservation and the high level of FAT conservation across the PIKK family, it is conceivable that ATM activation also involves this same mechanistic step, with the activator MRN-dsDNA complex triggering the Pincer to pull on and twist the FAT domain to realign the N and C lobes (*Figure 5—figure supplement 2B*).

If movement of the Pincer domain is on the pathway of ATM activation, we presume dsDNA and MRN binding will involve portions of ATM beyond the Spiral that binds to Nc28 and which has been proposed to have site(s) of dsDNA binding *Jansma et al., 2020*. The Spiral has a single, isolated interface with the Pincer domain, and it makes no other interactions to the remainder of the ATM dimer. As such, binding events that are limited to the Spiral would be unlikely to cause a movement or conformational change at the Pincer domain. With mTOR, the activator Rheb bridges one end of the N-heat solenoid to portions of mTOR that are invariant during activation, thus providing a pivot point for the movement of the other N-heat end anchored on the FAT domain (*Yang et al., 2017*). While DNA-PKcs is more complex, the extensive interactions of N-heat and M-heat along their solenoids appear to similarly serve as pivot point(s) for the movement of N-heat at its FAT anchor (*Chen et al., 2021b*). It is thus likely that dsDNA-MRN either engage additional ATM domains beyond the Spiral, or they bridge the two Spiral domains of the dimer in a manner that changes their relative orientation, with the resulting change propagating to the Pincer domains (*Figure 5—figure supplement 2B*).

Because the FAT domain plays a central role in the dimerization interface of the inactive ATM dimer, a conformational change in the FAT domain may well affect the relative arrangement of the two ATM protomers. This may disrupt the packing of kα9b with the TRD3 coiled coil from the second protomer, allowing the movement of kα9b to expose the $P_{+1}$ substrate-binding site. Our steady state kinetic analysis showed that MRN + dsDNA increases the $k_{cat}$ value of ATM phosphorylating a p53 substrate by over two orders of magnitude without affecting the $K_M$ value (*Figure 5—figure supplement 1C* to H). This mirrors the steady-state kinetic constants of mTOR activation (*Yang et al., 2017*). However, activation having no effect on the $K_M$ value was unexpected, given that the presumed binding site for the $P_{+1}$ position of the substrate is blocked in the structure. This may be analogous to findings with the canonical Cdk2-CyclinA kinase, which is activated by phosphorylation on its activation loop. Even though phosphorylation reorganizes the substrate-binding site on the Cdk2 activation loop, the majority of the increase in catalytic efficiency is reflected in an increased rate of phosphoryl group transfer step (*Hagopian et al., 2001*). Based on the model proposed for Cdk2 (*Hagopian et al., 2001*), it is possible that the blocked $P_{+1}$ position in inactive ATM allows substrate binding, but with the phospho-acceptor group in an unfavorable position or orientation that reduces the rate of phosphoryl group transfer.

In conclusion, our structural data supports the model that activation of the ATM kinase domain may be mechanistically analogous to the activation of mTOR and DNA-PKcs. While it is not known how Rad50 and Mre11 interact with ATM, we find that the Nbs1 FxF/Y motif binds to the ATM Spiral domain, and the Spiral domain of Tel1 was shown to bind to dsDNA independently of MRN (*Jansma et al., 2020*). Thus, it is possible that these Spiral interactions cooperate with additional contacts by MRN to trigger a motion of the Pincer domain. This would then result in a conformational change in the FAT domain that not only realigns the N and C lobes of the kinase domain but also relieves the blockage of the substrate-binding site.

## Materials and methods

### Peptides, reagents, and antibodies

A peptide corresponding to the C-terminal 28aa of Nbs1 (Nc28) was purchased from Genscript and dissolved in 100 mM HEPES pH 7.4 to a concentration of 2 mM as measured by 205 nm absorbance. No other specialty reagents or antibodies were used in this study.

### Purification of ATM kinase

Codon optimized human ATM harboring an N-terminal FLAG tag was purified from a stably transfected HEK 293 cell line grown in suspension. Cells were grown to a density of 2-3e$^6$ per liter, pelleted, and resuspended in 50 mL per liter of ATM lysis buffer containing 50 mM Tris-HCl pH 8.0, 500 mM NaCl, 50 mM KCl, 1 mM EDTA, 10 % v/v glycerol, 0.5 mM TCEP, supplemented with protease inhibitors aprotinin, leupeptin, pepstatin, AEBSF. Cells were lysed by two passages through a cell disruptor,

and the cell lysate was clarified by centrifugation. The soluble cell lysate was incubated with α-FLAG M2 sepharose (Sigma) for 1 hr at 4°C, passed over a gravity column, and the resin was extensively washed with lysis buffer. FLAG-ATM was eluted from the resin in lysis buffer containing 0.2 mg/mL FLAG peptide. FLAG-ATM was diluted to reduce the NaCl concentration to ~100 mM and loaded onto a Mono-Q 5/50 GL column equilibrated in Buffer A (25 mM Tris-Cl pH 8.0, 100 mM NaCl, and 0.5 mM TCEP). ATM was eluted from the column with a linear gradient of 0–100% buffer B (25 mM Tris-Cl pH 8.0, 1 M NaCl, and 0.5 mM TCEP). ATM eluted at approximately 325 mM NaCl. Peak fractions were pooled and used directly for cryo-EM grid preparation. The remaining fractions containing ATM were pooled and glycerol was added to a final concentration of 10 % v/v. ATM was aliquoted, flash frozen in liquid $N_2$, and stored at –80°C for kinase and pull-down assays.

## Purification of the MRN complex

Codon optimized human Mre11, Rad50, and Nbs1 each containing their own CMV promoters, non-cleavable C-terminal FLAG tag, and poly-A sequences were cloned into a modified pcDNA-based polycistronic mammalian expression vector. Mre11 also harbored the H129N mutation that abolishes nuclease activity. FLAG-tagged MRN was expressed by PEI transient transfection of HEK 293 cells growing in suspension at a density of 1-2e⁶ per liter. After 48 hrs, cells were pelleted and resuspended in 50 mL per liter of MRN lysis buffer containing 50 mM Tris-Cl pH 8.0, 750 mM NaCl, 50 mM KCl, 10 % v/v glycerol, 0.5 mM TCEP, supplemented with protease inhibitors aprotinin, leupeptin, pepstatin, AEBSF. The remainder of the purification procedure was performed using the same protocol as ATM. Peak Mono-Q fractions were pooled, aliquoted, flash frozen in liquid $N_2$, and stored at –80°C for kinase assays. The MRN FRY-to-AAA mutant was generated using InFusion mutagenesis and purified in the same manner as wild-type MRN.

## Cryo-EM grid preparation

For the apo ATM sample, Mono-Q purified ATM kinase was diluted to 0.6 mg/mL in 25 mM Tris-Cl pH 8.0, 300 mM NaCl, 0.5 mM TCEP, 1.25 mM AMP-PNP, and 2.5 mM $MgCl_2$ and incubated on ice for 1 hour. For the ATM-Nc28 sample, frozen ATM was thawed and dialyzed against 25 mM Tris-Cl pH 8.0, 250 mM NaCl, and 0.5 mM TCEP overnight. ATM was then mixed with the Nc28 peptide at final concentrations of 1.2 μM (0.42 mg/mL) and 207 μM (0.73 mg/mL), respectively. After addition of 2.5 mM AMP-PNP and 5 mM $MgCl_2$, the sample was incubated on ice for 1 hr. Each sample was briefly centrifuged to remove large aggregated species. The sample was applied to glow-discharged UltraAuFoil 300 mesh R1.2/1.3 grids (Quantifoil) via double-sided application of 2.5 + 2.5 μL. Grids were blotted for 1.5–3 s at 22° C and 95% relative humidity and plunge frozen in liquid ethane using a FEI Vitrobot Mark IV.

## Cryo-EM data collection

For the apo ATM sample, a single dataset was collected at the Memorial Sloan Kettering cryoEM Facility on a Titan Krios microscope operated at 300kEV equipped with a Gatan K3 Summit direct electron detector. Data was acquired using a defocus range of –0.6 to –1.6 μm and a pixel size of 1.056 Å. Each micrograph was acquired using a 3-s exposure and fractionated into 40 frames with a dose of 20 electrons per pixel per second. A total of 9028 micrographs were collected using 3 × 3 image shift. For the ATM-Nc28 sample, a single dataset was collected at Janelia Research Campus on a Titan Krios microscope operated at 300kEV equipped with a Gatan K3 Summit direct electron detector and an energy filter. Data was acquired using a defocus range of –1.0 to –2.5 μm and a pixel size of 1.078 Å. Each micrograph was acquired using a 3-s exposure and fractionated into 40 frames with a dose of 20 electrons per pixel per second. A total of 7,866 micrographs were collected using 3 × 3 image shift.

## Data processing and structure refinement

Beam-induced sample motions were corrected using MotionCorr2 software, and contrast transfer function parameters were estimated using CTFFIND4 software. All subsequent processing steps were performed using RELION-3 software. All reported resolutions are calculated from gold-standard refinement procedures with the FSC = 0.143 criterion after post-processing by applying a soft mask, correction for the modulation transfer function (MTF) of the detector, temperature-factor

sharpening, and correction of FSC curves to account for the effects of the soft mask as implemented in RELION. For apo-ATM, all processing steps are summarized in *Figure 1—figure supplements 2 and 3*. A total of 8,973 micrographs were selected based on a CTF estimated resolution cutoff of 10 Å. ~2.5 million particles were autopicked using 2D templates generated from a previously collected, smaller ATM dataset collected using a Gatan K2 Summit direct electron detector. Picked particles were cleaned by multiple rounds of binned and unbinned 2D and 3D classifications. The remaining ~1.5 million particles were subjected to two rounds of per-particle Bayesian training, polishing and CTF refinement taking per particle astigmatism and beam tilt into account. Additional rounds of 2D and 3D classifications were performed after polishing to remove truncated ATM dimers. The resulting 303,604 particles were refined while applying C2 symmetry to an overall resolution of 2.5 Å. An additional round of 3D classification using six classes was performed to subclassify the overall ATM dimer particles into open and closed states. The most open class containing 44,152 particles and the most closed class containing 44,114 particles were each refined while applying C2 symmetry to an overall resolution of 2.7 Å and 2.7 Å, respectively. For each particle set (overall, open, and closed), partial signal subtraction and symmetry expansion procedures were performed to align the ATM protomer using a soft mask generated from an initial build of the ATM protomer (*Scheres, 2016*). Refinements were performed using masks generated for the overall promoter, N-terminal solenoids (roughly corresponding to residues 1–1588), and FAT + Kinase domains including FAT-proximal portions of the N-terminal solenoids (roughly corresponding to residues 1588–3056). An initial model was docked into each focused map in Chimera software, and focused maps were combined using the combine_focused_maps tool in Phenix software with the local_weighting flag set to True and local_residues set to 10. The final overall, open, and closed protomers were iteratively built and refined against these composite maps using Coot and Phenix software. All structure and model-to-map validations were performed using Molprobity tools as implemented in Phenix software using composite protomer maps. The final overall, open, and closed dimers were generated by applying NCS operators calculated from the corresponding C2 symmetric dimer maps and refined into the corresponding dimer composite maps. For the data set of the ATM-Nc28 complex, all processing steps are summarized in *Figure 4—figure supplement 1*. A total of 6,531 micrographs were selected based on a CTF estimated resolution cutoff of 5 Å. ~2 million particles were autopicked using 2D templates generated from a previous ATM dataset collected using a Gatan K2 Summit direct electron detector. Picked particles were cleaned by multiple rounds of binned and unbinned 2D and 3D classifications, with truncated ATM dimers removed prior to polishing. The remaining 287,615 particles were subjected to 2 rounds of per-particle Bayesian training, polishing and CTF refinement also taking per particle astigmatism and beam tilt into account. The polished particles were refined while applying C2 symmetry to an overall resolution of 2.5 Å. Partial signal subtraction, symmetry expansion, and focused refinement procedures were performed in the same manner as unbound ATM. Focused maps were combined in the same manner as unbound ATM, and an initial model was refined against the composite map. At this stage, we identified extra density adjacent to a hydrophobic cleft in the Spiral domain of ATM. A 7-residue segment ($^{742}$DLFRYNP$^{748}$) of the Nbs1 peptide had unambiguous density, especially for Phe744, Tyr746, and Pro748, and it was built into the map. Because the peptide density level was lower than the surrounding ATM density, we performed 3D classification without alignment to separate Nc28-bound and unbound particles using RELION. For this, we used the initial Nc28 model to generate a soft mask around this area including ATM residues 917–1101, and we performed iterative rounds of focused 3D classification without alignment using particles pre-aligned on the Spiral domain, with three classes and tau_fudge factor set to between 150–600. After multiple rounds of 3D local classification, one class containing ~75% of the original symmetry-expanded set of particles (481,066 particles) had strong density within the pocket. Focused refinements were repeated and maps were again combined using this subset of particles. The peptide density was better defined in the resulting composite map and we were able to extended the Nbs1 model to 10-residues ($^{740}$ADDLFRYNPY$^{749}$). The final protomer was built and refined against this composite protomer map using Coot and Phenix software. All structure and model-to-map validations were performed using Molprobity tools as implemented in Phenix software. The final ATM-Nc28 dimer was generated by applying NCS operators calculated from the C2 symmetric dimer map and refined into the dimer composite map.

## Pull-down assays

GST-tagged Nbs1 C-terminal 147 residues (GST-Nc147) harboring a C-terminal StrepII tag was expressed from a pGEX vector in *E. coli* BL21[DE3] cells. Cells were lysed by sonication and the protein was initially purified by a glutathione sepharose column. Elutions were further purified by a StrepTactin Sepharose column and eluted with 10 mM desthiobiotin (DTB). GST-Nc147 and mutants were concentrated and DTB was removed by serial buffer exchanges. Proteins were aliquoted, flash frozen in liquid N$_2$, and stored at –80°C. For pull-down assays, 3 µg of wild-type or mutant GST-Nc147 was coupled to 25 µL of pre-equilibrated glutathione sepharose resin in 200 µL of binding/wash (B/W) buffer containing 25 mM Tris-Cl pH 8.0, 200 mM NaCl, 0.5 mM TCEP, and 0.01 % v/v NP-40 for 1 hr with rotation at 4°C. The resin was washed once with 500 µL of B/W buffer, and resuspended in 200 µL of B/W buffer containing 0.05 mg/mL of FLAG-ATM (10 µg per reaction) and incubated for 2 hr rotating at 4°C. The resin was washed three times with 500 µL of B/W buffer for 10 min rotating at 4°C. Twenty-five µL of pelleted resin was resuspended in 25 µL of 2 X LDS sample buffer and boiled for 10 min. A total of 25 µL (50% of the reaction) was loaded onto a 4–12% Bis-Tris SDS-PAGE gel. Proteins were separated by electrophoresis and gels were stained in InstantBlue Coomassie Stain (Abcam).

## In-vitro kinase assays

All steady-state in-vitro kinase assays were assembled in 10 µL in a buffer containing 25 mM HEPES-Na pH 7.5, 150 mM NaCl, 1 mM MgCl$_2$, 2 mM DTT, 5 % v/v glycerol, and 1 mg/mL native BSA. All kinase assays also contained 25 nM ATM kinase. The substrate used for all kinase assays is a GST-tagged p53 fragment of 44 N-terminal residues (GST-p53 residues 1–44). Reactions were assembled on ice, and started by the addition of 0.5 mM cold ATP supplemented with 4 µCi [γ-$^{32}$P] ATP (6000 Ci/mmol, Perkin-Elmer) per reaction. Reactions proceeded for 1 hr at 30°C, and were stopped by the addition of 20 µL of stop buffer containing 1.5 X NuPAGE LDS sample buffer, 40 mM EDTA, and 30 mM TCEP. Ten µL (1/3 of the reaction) was loaded onto a 4–12% Bis-Tris SDS-PAGE gel and proteins were separated by electrophoresis. ATP standards were spotted and gels were dried on DE-81 ion exchange cellulose paper. Dried gels were exposed to a phosphor imaging plate and imaged on a Typhoon FLA 7000 imager. Bands were quantified using ImageJ software and the concentrations of phosphorylated product and enzymatic velocities were calculated based on spotted ATP standards. Velocities were normalized to the concentration of ATM (V/[E] (s$^{-1}$)) and enzymatic parameters were calculated by fitting the data to the Michaelis-Menten equation:

$$Y = \frac{k_{cat} \times [S]}{k_M \times [S]}$$

For MRN/dsDNA titrations and EC$_{50}$ calculations (*Figure 5A-C*, *Figure 4—figure supplement 2C*, *Figure 5—figure supplement 1B and D*) the substrate was kept fixed at 25 µM concentration. EC$_{50}$ values, hill coefficients (*n*), and fold-activation were calculated by fitting the data to the following '[agonist] vs. response (four parameters)' equation in Prism software:

$$Y = \frac{Bottom + [activator]^n \times (Top - Bottom)}{[activator]^n + EC_{50}^n}$$

For single-turnover assays (*Figure 5—figure supplement 1G*), large reactions were assembled containing 250 nM ATM + 25 nM 250 bp dsDNA ±250 nM MRN ± 100 nM p53 substrate and incubated at 25°C for 5 min. ATP containing 4 µCi [γ-$^{32}$P] ATP was added at a final concentration of 0.5 mM, and reactions were incubated at 25°C. Ten µL aliquots were removed at specified time points and mixed with 20 µL of stop buffer. The remainder of the assay was performed in the same manner as steady-state kinase assays.

## Acknowledgements

We thank Haijuan Yang for establishing the FLAG-tagged ATM HEK 293 stable cell line and establishing preliminary purification procedures. We also thank Jason De La Cruz and Doreen Matthies for assistance in data collection at the MSK and HHMI cryoEM facilities, respectively. This work was supported by HHMI, NIH grant CA008748 and NCI training grant 5F32CA247320 (to C.W.).

## Additional information

### Funding

| Funder | Grant reference number | Author |
|---|---|---|
| National Cancer Institute | 5F32CA247320 | Christopher Warren |
| National Cancer Institute | CA008748 | Nikola P Pavletich |

The funders had no role in study design, data collection and interpretation, or the decision to submit the work for publication.

### Author contributions

Christopher Warren, Formal analysis, Funding acquisition, Investigation, Methodology, Validation, Writing - original draft, Writing – review and editing; Nikola P Pavletich, Conceptualization, Funding acquisition, Supervision, Validation, Writing – review and editing

### Author ORCIDs

Christopher Warren (iD) http://orcid.org/0000-0002-0320-7248
Nikola P Pavletich (iD) http://orcid.org/0000-0002-6039-3956

### Decision letter and Author response

Decision letter https://doi.org/10.7554/eLife.74218.sa1
Author response https://doi.org/10.7554/eLife.74218.sa2

## Additional files

### Supplementary files

- Supplementary file 1. Cryo-EM data processing and refinement table.
- Transparent reporting form
- Source data 1. Source data - original gel images.

### Data availability

The refined dimer structures and corresponding cryo-EM maps (including consensus, symmetry expanded, focused, and composite maps) for unbound and Nc28-bound ATM have been deposited within the Protein Data Bank (PDB) and the Electron Microscopy Data Bank (EMDB) under accession codes 7SIC and EMD-25140 (unbound) and 7SID and EMD-25141 (Nc28-bound).

The following datasets were generated:

| Author(s) | Year | Dataset title | Dataset URL | Database and Identifier |
|---|---|---|---|---|
| Warren C, Pavletich NP | 2022 | Structure of the human ATM kinase and mechanism of Nbs1 binding | https://www.rcsb.org/structure/7SIC | RCSB Protein Data Bank, 7SIC |
| Warren C, Pavletich NP | 2022 | Structure of the human ATM kinase and mechanism of Nbs1 binding | https://www.rcsb.org/structure/7SID | RCSB Protein Data Bank, 7SID |
| Warren C, Pavletich NP | 2022 | Structure of the human ATM kinase and mechanism of Nbs1 binding | http://www.ebi.ac.uk/pdbe/entry/emdb/EMD-25140 | Electron Microscopy Data Bank, EMD-25140 |
| Warren C, Pavletich NP | 2022 | Structure of the human ATM kinase and mechanism of Nbs1 binding | http://www.ebi.ac.uk/pdbe/entry/emdb/EMD-25141 | Electron Microscopy Data Bank, EMD-25141 |

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
