## [Editor Report]

This manuscript is of broad interest to the DNA-repair and structural biology field. The paper describes new insights into the interaction between ATM and Nsb1, proteins central to repairing DNA double-strand breaks in humans. Overall, the structural cryo-electron microscopy data is solid and the data well analyzed and presented with key claims directly related to and supporting previous known findings.

---

## [Decision Letter]

**Decision letter after peer review:**

Thank you for submitting your article "Structure of the human ATM kinase and mechanism of Nbs1 binding" for consideration by *eLife*. Your article has been reviewed by 3 peer reviewers, and the evaluation has been overseen by a Reviewing Editor and Philip Cole as the Senior Editor. The reviewers have opted to remain anonymous.

Essential revisions:

1) The kinase reactions are missing an important control: Please provide a phosphorylation reaction with DNA, MRN and no ATM. This will exclude that MRN co-purified with a contaminating kinase (as noted e.g. by the Burgers laboratory in yeast).

2) Since ATM has structural analogies to ATR and PRD is important for ATR activation, the authors could further discuss the PRD activation in ATM, e.g., the equivalent mutant (K3016 in ATM?) of K2589E in ATR, or others? There are also evident structural analogies with DNA-PKcs. On the most basic scale there appear to be parallels between the DNA binding sensor and the flexible attachment to the PIKK could the authors discuss this in the paper for readers striving to understand how this important new structure impacts understanding for this key kinase family?

3) Although disruption of the Nbs1 peptide binding site does not seem to greatly affect ATM activation by MRN, could the authors discuss and/or test if any of the cancer mutations around the Nbs1 binding site impact ATM activity? Here background discussion could be strengthened. Substantial structural and biological information exists for MRN, so readers might benefit from citing a suitable review on this dynamic complex (e.g. PMID: 29709199 DOI: 10.1146/annurev-biochem-062917-012415) and also on the biochemical and biological results on ATM activation (e.g. PMID: 25580527 DOI: 10.1146/annurev-biochem-060614-034335). The Nbs1 structured domains and C-terminal disorder were originally experimentally defined in the Nbs1 structure paper by combined X-ray crystallography and X-ray scattering, and this could be cited (PMID: 19804755 DOI: 10.1016/j.cell.2009.07.033) and discussed in the Nbs1 peptide choice.

4) Comment on the local resolution in the region where the peptide has been built, although overall 2.6 A resolution from Figure S9 that region looks closer to 4.7-5 A?

Comment on the ease and confidence of building the peptide, it looks like it was based on density for the Phe and Tyr? It also looks like from opening the map and pdb in coot and chimera that the peptide could fit into the density better – Arg 745 and Pro 748 need to be moved into the density more – you can see this is Figure 4C (right).

Comment on the percentage of particles which went into the peptide bound structure, did you observe any without peptide bound in this dataset?

Add cryo-EM view distribution to show preferred orientation – this is now standard for cryo-EM data. 262 (page 13): typo: would the main mechanism

5) ATM activation by MRN requires long DNA (>200 bp). From the ATM electrostatic surface, it does not seem to have many regions with positive charges except the tip of Spiral domain. Could authors discuss their view on why long DNA may be required?

6) It would be helpful to ongoing efforts if the authors could propose a testable working model and figure for ATM activation to help understand the structural implications thought a visual explanation.

7) It is correct that ref23 indicates that yeast Mre11-Rad50 can activate Tel1 without Xrs2; however, this is rather a minority observation. Maybe the authors could cite two papers from the Symington lab (Oh et al., 2016 and 2018), which nicely link the activation to Xrs2.

8) It would be helpful to ongoing efforts if the authors could propose a testable working model and figure for ATM activation to help understand the structural implications thought a visual explanation. The data in S11E suggest that MRN does not recruit ATM to DNA, but rather that MRN activates is kinase activity, which is in contrast to models to date. While the observation may be specific to the biochemical setup, the authors may want to discuss these striking data more and add the limitation of their conclusions.

---

## [Author Response]

Essential revisions:1) The kinase reactions are missing an important control: Please provide a phosphorylation reaction with DNA, MRN and no ATM. This will exclude that MRN co-purified with a contaminating kinase (as noted e.g. by the Burgers laboratory in yeast).

We now include this control as a new Figure 5**—**supplemental figure1C and legend (line 1187). This assay was repeated 3 times with similar results.

2) Since ATM has structural analogies to ATR and PRD is important for ATR activation, the authors could further discuss the PRD activation in ATM, e.g., the equivalent mutant (K3016 in ATM?) of K2589E in ATR, or others?

We now discuss the ATR K2589E mutation and the corresponding E3007 of ATM (Line 464):

“While scanning mutagenesis identified an ATR kα10 residue that disrupts TopBP1-mediated activation^60^, it is not clear what role kα10 plays in PIKK activation. The mutation (K2589E) maps to a surface residue uninvolved in any interactions in the inactive ATR structure^61^, and the corresponding ATM residue (Glu3007) is similarly surface-exposed and devoid of any interactions. In addition, the kα10 helix is an integral part of the PIKK C lobe structure, and it is structurally invariant among inactive and active PIKK structures.”

(K3016 in ATM?)

K3016 does not correspond to K2589E in ATR (K3016 is making buried hydrogen bonds to E2895 and CO of Y2864, and it has a structural role).

, or others?

The other ATR PRD mutation that affects ATR activation in the Mordes et al. 2008 paper is HVL2591AAA at kα10 residues that have structural roles. The rest of the mutations, including K2589A had no effect.

There are also evident structural analogies with DNA-PKcs. On the most basic scale there appear to be parallels between the DNA binding sensor and the flexible attachment to the PIKK could the authors discuss this in the paper for readers striving to understand how this important new structure impacts understanding for this key kinase family?

We have expanded our discussion of the similarities with DNA-PKcs (line 507):

“DNA-PKcs activation is triggered by the N-heat solenoid binding to the dsDNA end, stabilized by the Ku70-Ku80 complex. The Ku complex interacts with DNA and the N- and M-heat DNA-PKcs domains, but it also utilizes two flexibly linked elements for additional contacts to DNA-PKcs. These may reflect an initial recruitment interaction, a role that the Nbs1 Nc28 peptide may also be involved in (Figure 4—figure supplement 2E).”

We have also expanded the discussion of DNA-PKcs activation and its relevance to ATM activation (line 537):

“While DNA-PKcs is more complex, the extensive interactions of N-heat and M-heat along their solenoids appear to similarly serve as pivot point(s) for the movement of N-heat at its FAT anchor44. It is thus likely that dsDNA-MRN either engage additional ATM domains beyond the Spiral, or they bridge the two Spiral domains of the dimer in a manner that changes their relative orientation, with the resulting change propagating to the Pincer domains (Figure 5—figure supplement 2B).”

3) Although disruption of the Nbs1 peptide binding site does not seem to greatly affect ATM activation by MRN, could the authors discuss and/or test if any of the cancer mutations around the Nbs1 binding site impact ATM activity?

We have expanded the discussion of ATM mutations at the Nbs1 binding site (line 324):

“… and the structure suggests that the S978P mutation would disrupt the sα39 helix that forms part of the Nbs1-binding site, while S978A and S978Y would eliminate the interactions of Ser978 with Asp741 of Nbs1 (Figure S5D and E). Other cancer-associated missense mutations at the Nbs1 binding groove occur at lower frequencies. The structure suggests that the S974F mutation, in a region of limited solvent accessibility due to the Nc28 peptide, would likely cause a steric clash either with Arg745 of Nbs1 or other ATM backbone and side chain groups; R981C/H would eliminate the contact to Asp 741 of Nbs1; C987Y/W and F1025S/L, which map to a local hydrophobic core behind the sα39 and sα42 helices, would disrupt the structural integrity of the binding site of Nbs1 Phe744 and its vicinity.”

Here background discussion could be strengthened. Substantial structural and biological information exists for MRN, so readers might benefit from citing a suitable review on this dynamic complex (e.g. PMID: 29709199 DOI: 10.1146/annurev-biochem-062917-012415) and also on the biochemical and biological results on ATM activation (e.g. PMID: 25580527 DOI: 10.1146/annurev-biochem-060614-034335).

We have included the references with *PMID: 29709199* at line 37 and *PMID: 25580527* at line 41 in the Introduction.

The Nbs1 structured domains and C-terminal disorder were originally experimentally defined in the Nbs1 structure paper by combined X-ray crystallography and X-ray scattering, and this could be cited (PMID: 19804755 DOI: 10.1016/j.cell.2009.07.033) and discussed in the Nbs1 peptide choice.

We have cited the reference and rationale for using a peptide at line 283:

“As the Nbs1 C-terminal half is unstructured or loosely folded^23^, we made cryo-EM grids using a 28-residue peptide…”.

4) Comment on the local resolution in the region where the peptide has been built, although overall 2.6 A resolution from Figure S9 that region looks closer to 4.7-5 A?Comment on the ease and confidence of building the peptide, it looks like it was based on density for the Phe and Tyr?

The consensus reconstruction in Figure 4—figure supplement 1C has lower resolution in the Spiral-peptide region than the reconstruction that was refined with a mask focused on the Spiral+peptide, for which we now provide the local resolution estimation, including a close-up of the peptide region, in Figure 4—figure supplement 1G. The figure shows that the central portion of Nc28 has a resolution better than ~3.5 Å. We also expand on it in the main text (line 295):

“…guided by the unambiguous density of the side chains for Phe744 and Tyr746. According to local resolution estimation, the focused reconstruction has a resolution better than ~3.5 Å in the central portions of the peptide (Figure 4B, Figure 4—figure supplement 1G).”.

It also looks like from opening the map and pdb in coot and chimera that the peptide could fit into the density better – Arg 745 and Pro 748 need to be moved into the density more – you can see this is Figure 4C (right).

We have further refined the peptide, and we have updated Figure 4C and related peptide figures with the improved model.

Comment on the percentage of particles which went into the peptide bound structure, did you observe any without peptide bound in this dataset?

The percentage of particles containing the peptide was stated to be 75 % in the text/methods. We now also provide the number of particles (481,066 particles, lines 292 and 694).

Add cryo-EM view distribution to show preferred orientation – this is now standard for cryo-EM data. 262 (page 13): typo: would the main mechanism

We now provide a cryo-EM view distribution Figure 1—figure supplement 2E and Figure 4—figure supplement 1F, for apo- and Nc28-bound ATM, respectively. We have corrected the typo (now line 257).

5) ATM activation by MRN requires long DNA (>200 bp). From the ATM electrostatic surface, it does not seem to have many regions with positive charges except the tip of Spiral domain. Could authors discuss their view on why long DNA may be required?

The possible explanations for the long DNA requirement include “…such as the ability to span an extended binding site across the ATM dimer or to link distal binding sites on ATM-MRN, may be important” (line 412). These possibilities are difficult to deconvolute as both ATM and MRN have been shown to independently bind to dsDNA. We do not mention other possibilities, such as linking two or more ATM dimers, for which there is little experimental support.

6) It would be helpful to ongoing efforts if the authors could propose a testable working model and figure for ATM activation to help understand the structural implications thought a visual explanation.

We provide a Figure 5—figure supplement 2B with a model and have expanded the discussion of a working model of ATM activation (lines 529 to 542):

“If movement of the Pincer domain is on the pathway of ATM activation, we presume dsDNA and MRN binding will involve portions of ATM beyond the Spiral that binds to Nc28 and which has been proposed to have site(s) of dsDNA binding^39^. The Spiral has a single, isolated interface with the Pincer domain, and it makes no other interactions to the remainder of the ATM dimer. As such, binding events that are limited to the Spiral would be unlikely to cause a movement or conformational change at the Pincer domain. With mTOR, the activator Rheb bridges one end of the N-heat solenoid to portions of mTOR that are invariant during activation, thus providing a pivot point for the movement of the other N-heat end anchored on the FAT domain^42^. While DNA-PKcs is more complex, the extensive interactions of N-heat and M-heat along their solenoids appear to similarly serve as pivot point(s) for the movement of N-heat at its FAT anchor^44^. It is thus likely that dsDNA-MRN either engage additional ATM domains beyond the Spiral, or they bridge the two Spiral domains of the dimer in a manner that changes their relative orientation, with the resulting change propagating to the Pincer domains (Figure 5—figure supplement 2B).”

7) It is correct that ref23 indicates that yeast Mre11-Rad50 can activate Tel1 without Xrs2; however, this is rather a minority observation. Maybe the authors could cite two papers from the Symington lab (Oh et al., 2016 and 2018), which nicely link the activation to Xrs2.

We have expanded on this aspect in the Introduction and cited the two references (lines 57-60):

“In the yeast system, Xrs2 is required for Tel1 recruitment to DSBs in vivo^27,28^, but in vitro Tel1 activation appears more dependent on Rad50, as either Rad50-Mre11 or Rad50-Xrs2 but not Mre11-Xrs2 can partially activate Tel1 in the presence of DNA^26^”.

8) It would be helpful to ongoing efforts if the authors could propose a testable working model and figure for ATM activation to help understand the structural implications thought a visual explanation.

We have provided Figure 5—figure supplement 2B and expanded the discussion of ATM activation (lines 529 to 542) in response to comment 6 above.

The data in S11E suggest that MRN does not recruit ATM to DNA, but rather that MRN activates is kinase activity, which is in contrast to models to date. While the observation may be specific to the biochemical setup, the authors may want to discuss these striking data more and add the limitation of their conclusions.

We now stress that our observations may be specific for our biochemical setup, and they may not reflect the MRN requirement for recruiting ATM to DSBs (lines 431-434):

“We note, however, that our in vitro assay conditions, such as the saturating dsDNA concentration, may not recapitulate the demonstrated requirement for MRN recruiting ATM to DNA in vivo^27–29^.